# Peri active site catalysis of proline isomerisation is the molecular basis of allomorphy in β-phosphoglucomutase
F. Aaron Cruz-Navarrete [1,3,6], Nicola J. Baxter [1,2,6], Adam J. Flinders[1,4], Anamaria Buzoianu[1,5], Matthew J. Cliff [2], Patrick J. Baker [1] & Jonathan P. Waltho [1,2] ✉

Metabolic regulation occurs through precise control of enzyme activity. Allomorphy is a post-translational fine control mechanism where the catalytic rate is governed by a conformational switch that shifts the enzyme population between forms with different activities. β-Phosphoglucomutase (βPGM) uses allomorphy in the catalysis of isomerisation of β-glucose 1-phosphate to glucose 6-phosphate via β-glucose 1,6-bisphosphate. Herein, we describe structural and biophysical approaches to reveal its allomorphic regulatory mechanism. Binding of the full allomorphic activator β-glucose 1,6-bisphosphate stimulates enzyme closure, progressing through NAC I and NAC III conformers. Prior to phosphoryl transfer, loops positioned on the cap and core domains are brought into close proximity, modulating the environment of a key proline residue. Hence accelerated isomerisation, likely via a twisted *anti*/C4-*endo* transition state, leads to the rapid predominance of active *cis*-P βPGM. In contrast, binding of the partial allomorphic activator fructose 1,6-bisphosphate arrests βPGM at a NAC I conformation and phosphoryl transfer to both *cis*-P βPGM and *trans*-P βPGM occurs slowly. Thus, allomorphy allows a rapid response to changes in food supply while not otherwise impacting substantially on levels of important metabolites.

The control of metabolism in all living systems is achieved through precise regulation of enzyme activity[1–3] and failures of these regulation mechanisms often result in metabolic diseases and disorders[4,5]. Post-translational fine control mechanisms, such as reversible covalent modifications, manipulate enzyme activity over short timescales (millisecond to minutes)[6,7]. Further fine-control can be provided by allostery, allokairy and allomorphy mechanisms, where the overall catalytic rate is governed by a conformational switch that shifts the enzyme population between forms that have different activities. Allostery operates via an allosteric effector molecule binding somewhere other than the active site, which stabilises forms of the enzyme with either an enhanced or reduced activity[2,8–10]. Allokairy uses the binding of the substrate in the active site to shift the enzyme population from a low activity form to a fully active form at a conformational exchange rate that is similar to the rate of catalysis[11,12]. Allomorphy is similar to allokairy but involves the binding of an activator molecule in the active site to shift the enzyme population to a fully active form and exploits a conformational exchange process that is much slower than the rate of catalysis[13].

Such enzyme control mechanisms allow organisms to adapt their metabolism in response to changing environmental conditions and to exploit different energy sources. Recently, β-phosphoglucomutase (βPGM, EC 5.4.2.6, 25 kDa) has been identified as the first example of an allomorphic enzyme. Here, a substantial part of the basal population is maintained in a more latent state that can be converted rapidly (~seconds) into the most active form when required[13]. This fine control mechanism provides *Lactococcus lactis* (*L. lactis*) with a means to respond quickly to changes in carbohydrate sources, while minimising the unproductive diversion of valuable metabolites[14–19]. βPGM is a monomeric, magnesium-dependent phosphoryl transfer enzyme belonging to the haloacid dehalogenase superfamily of phosphomutases[13,20–29]. During typical steady-state catalysis, β-glucose 1-phosphate (βG1P) binds to phosphorylated βPGM

[1]School of Biosciences, University of Sheffield, Sheffield, S10 2TN, UK. [2]Manchester Institute of Biotechnology and Department of Chemistry, The University of Manchester, Manchester, M1 7DN, UK. [3]Present address: Department of Structural Biology, St. Jude Children's Research Hospital, Memphis, TN, 38105, USA. [4]Present address: Cancer Research UK, Manchester Institute, Patterson Building, Manchester, M20 4BX, UK. [5]Present address: Department of Chemistry, Biochemistry and Pharmacy, University of Bern, Bern, 3012, Switzerland. [6]These authors contributed equally: F. Aaron Cruz-Navarrete, Nicola J. Baxter. ✉e-mail: j.waltho@sheffield.ac.uk

($\beta PGM^P$, phosphorylated at residue D8) generating a transient β-glucose 1,6-bisphosphate (βG16BP) reaction intermediate. Release to solution and rebinding to substrate-free βPGM in an alternative orientation[23], results in the formation of glucose 6-phosphate (G6P) and regeneration of $\beta PGM^P$ (Supplementary Fig. 1a). Substrate-free βPGM exists as two distinct conformers with different activities due to *cis-trans* isomerisation of the K145-P146 peptide bond on the multi-second timescale[13]. Fully active βPGM has P146 in the *cis* form (*cis*-P βPGM) and the K145 sidechain is engaged in the active site (PDB 1ZOL[21], PDB 2WHE[26]). In contrast, when P146 is in the *trans* form (*trans*-P βPGM), the K145 sidechain is repositioned away from the active site and is exposed to solvent (PDB 6YDK[13]).

The lifetime of $\beta PGM^P$ is short (~30 s in vitro[25]) and therefore substrate-free βPGM requires the action of a phosphorylating agent to form $\beta PGM^P$. βG16BP not only phosphorylates βPGM, but is a very effective allomorphic activator of the enzyme, resulting rapidly in the full population of *cis*-P $\beta PGM^P$ (Fig. 1b, Supplementary Fig. 1b), i.e. βG16BP is a full allomorphic activator[13]. In the absence of βG16BP in vivo, the most abundant phosphorylating agent is predicted to be the glycolytic intermediate fructose 1,6-bisphosphate (F16BP), which can reach intracellular concentrations of ~50 mM when *L. lactis* is grown in glucose-rich media[17]. When F16BP is used as a phosphorylating agent in vitro[13], both *cis*-P $\beta PGM^P$ and *trans*-P $\beta PGM^P$ are produced. The presence of both species results in the observation of a pronounced lag phase in activity until the population of *cis*-P $\beta PGM^P$ dominates (Fig. 1a, Supplementary Fig. 1b), i.e. F16BP is a partial allomorphic activator.

The active site of βPGM is located in a cleft formed between the cap domain (T16–V87) and the core domain (M1–D15, S88–K221), and these domains reorient in response to the binding of a phosphodianion group in the distal region of the active site[30] (Fig. 2). Catalysis requires exchange between the open substrate-free conformation and the fully closed near-transition state conformation[21,22,26]. Two other discrete states have been identified within the catalytic cycle and both correspond to a near-attack conformer (NAC)[31,32]. The NAC I conformation (PDB 2WF9[27]) has an interdomain hinge closure angle of ~26° when compared with the open substrate-free conformation, whereas for the NAC III conformation (PDB 5OK1[29]) this angle is ~35° (Fig. 1c–e, Supplementary Table 1). The NAC III conformation is adopted by the D10N variant ($\beta PGM_{D10N}$) when binding βG16BP (*cis*-P $\beta PGM_{D10N}$:βG16BP complex), where the carboxylate to carboxamide sidechain substitution of the general acid-base residue mimics a protonated D10 sidechain, but is unable to transfer a proton to the nascent leaving group[29,32].

The underlying structural mechanisms that produce allomorphic control of βPGM have not been described. However, the binding properties of $\beta PGM_{D10N}$ provide an elegant approach to investigate the differences between partial allomorphic activation by F16BP and full allomorphic activation by βG16BP. Here we show, through a combined use of site-directed mutagenesis, X-ray crystallography and NMR spectroscopy that the allomorphic regulatory mechanism operating in βPGM is delivered by the action of a substrate specificity loop (V36–L53[33]). Its proximity to an allomorphic control loop (E140–I150) influences the proficiency of *trans* to *cis* isomerisation of the K145-P146 peptide bond located at the periphery of the active site (Fig. 2). On binding F16BP, the usual progression between conformations induced by substrate binding is stalled at NAC I by steric clashes imposed between the substrate specificity loop and two fructose ring hydroxyl groups. Consequently, both *trans*-P βPGM and *cis*-P βPGM remain populated and therefore the overall catalytic activity is low. On

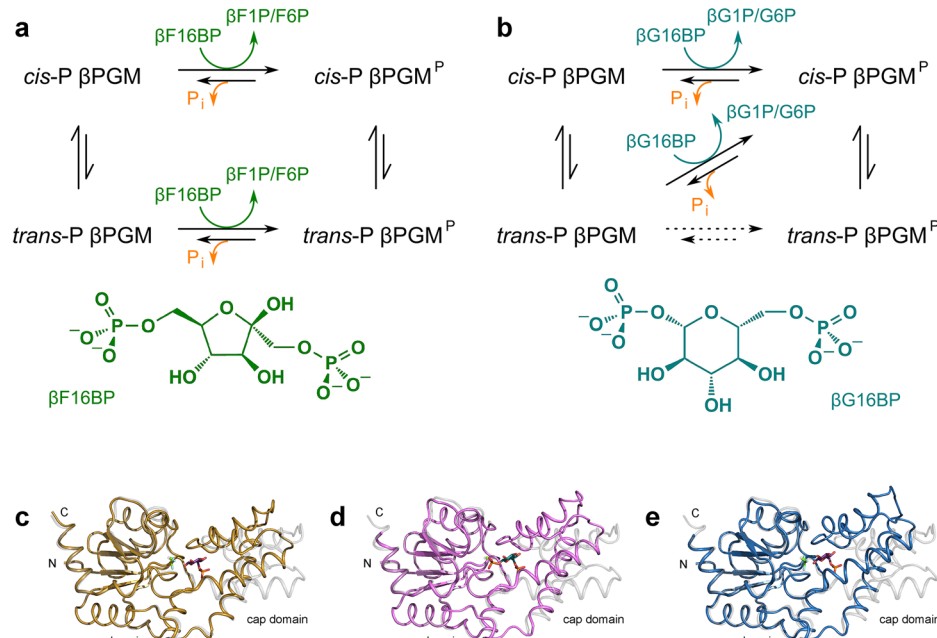

**Fig. 1 | Reaction schemes for the allomorphic activation of βPGM and differences in the interdomain hinge closure angle for βPGM structures. a, b** Reaction schemes for the phosphorylation of βPGM using either F16BP or βG16BP as allomorphic activators. Substrate-free βPGM exists as two conformers with different activities that result from *cis-trans* isomerisation of the K145-P146 peptide bond[13]. The fully active form is *cis*-P βPGM. **a** Both *cis*-P βPGM and *trans*-P βPGM are phosphorylated by F16BP (as βF16BP) generating *cis*-P $\beta PGM^P$ and *trans*-P $\beta PGM^P$, together with the release of either β-fructose 1-phosphate (βF1P) or fructose 6-phosphate (F6P). A pronounced lag phase is observed in catalytic activity until the population of *cis*-P $\beta PGM^P$ dominates. **b** The βG16BP reaction intermediate is able to couple the conformational switch and the phosphorylation step, resulting in the rapid generation of *cis*-P $\beta PGM^P$ along with either βG1P or G6P as products. A linear, fast initial rate is observed in kinetic profiles. Although the *trans*-P βPGM to *trans*-P $\beta PGM^P$ phosphorylation reaction (long dotted arrow) is possible, it has not been observed experimentally. Hydrolysis reactions, liberating inorganic phosphate ($P_i$), yield short lifetimes for both *cis*-P $\beta PGM^P$ and *trans*-P $\beta PGM^P$ (~30 s in vitro[25]). **c–e** Differences in the interdomain hinge closure angle for βPGM. Core domain superposition of the open substrate-free conformation (pale grey ribbon, *cis*-P $\beta PGM_{WT}$, PDB 2WHE[26]) with either (**c**) the NAC I conformation (gold ribbon, *cis*-P $\beta PGM_{WT}$:BeF3:G6P complex, PDB 2WF9[27]), (**d**) the NAC III conformation (pink ribbon, *cis*-P $\beta PGM_{D10N}$:βG16BP complex, PDB 5OK1[29]) or (**e**) the fully closed near-transition state conformation (blue ribbon, *cis*-P $\beta PGM_{WT}$:MgF3:G6P complex, PDB 2WF5[26]). $Mg_{cat}$ is depicted as a green sphere, G6P is shown as purple sticks, βG16BP is shown as teal sticks, and the $BeF_3^-$ and $MgF_3^-$ moieties are illustrated as green and pale blue sticks.

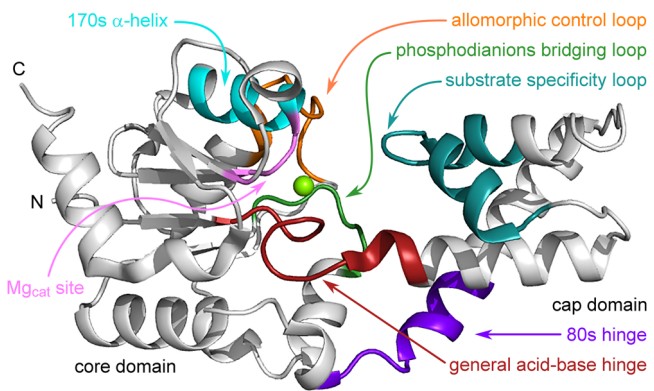

**Fig. 2 | βPGM enzyme architecture.** Cartoon representation of substrate-free *cis*-P βPGM (PDB 2WHE[26]) highlighting the architecture of the helical cap domain (T16–V87) and the α/β core domain (M1–D15 and S88–K221). The active site is located in the cleft formed between the domains and rotation at the hinge results in closure of the active site during catalysis. Key functional loops and structural motifs are indicated: general acid-base hinge (dark red, F7–E18), substrate specificity loop (teal, V36–L53), 80 s hinge (purple, N79–S88), phosphodianions bridging loop (green, A113–N118), allomorphic control loop (orange, E140–I150), Mg$_{cat}$ site (pink, E169–S171, V188) and 170 s α-helix (cyan, Q172–K179). The location of Mg$_{cat}$ is shown by a green sphere.

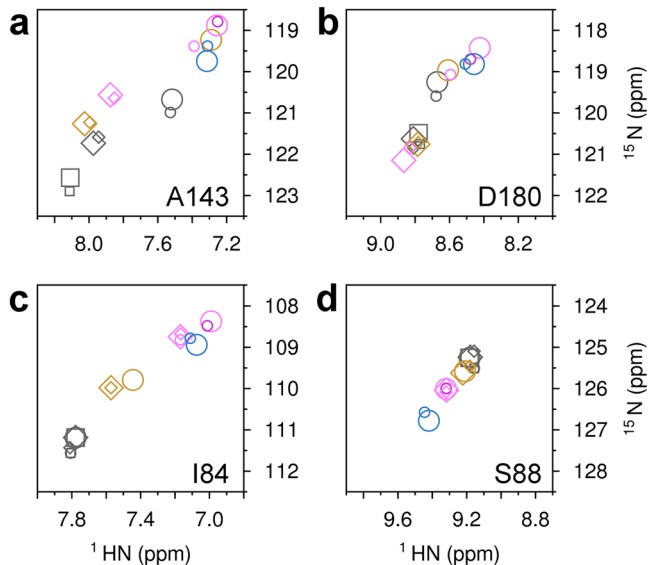

**Fig. 3 | $^1$H$^{15}$N-TROSY chemical shift comparisons of substrate-free βPGM species and βPGM complexes. a, b** Due to their structural proximity, residues A143 and D180 are sensitive reporters of the isomerisation state of the K145-X146 peptide bond. *trans*-X βPGM species (diamonds and squares) and *cis*-X βPGM species (circles) cluster in separate regions of their respective $^1$H$^{15}$N-TROSY spectra. **c, d** Residues I84 and S88 are located within the 80 s hinge and are reporters of the interdomain hinge closure angle. The open substrate-free conformation (black symbols), along with the NAC I conformation (gold symbols) and NAC III conformation (pink and purple symbols) describe a transition in both $^1$HN and $^{15}$N chemical shifts towards the fully closed near-transition state conformation (blue symbols). For I84, NAC III conformations are described by $^1$HN ~ 7.00 ppm, whereas NAC III$^t$ conformations are described by $^1$HN ~ 7.17 ppm. The βPGM species compared are: substrate-free *cis*-P βPGM$_{WT}$ (large black circles, BMRB 28095[13]), substrate-free *trans*-P βPGM$_{WT}$ (large black squares, BMRB 28096[13]), substrate-free *trans*-A βPGM$_{P146A}$ (large black diamonds, BMRB 27920[34]), substrate-free *cis*-P βPGM$_{D10N}$ (small black circles), substrate-free *trans*-P βPGM$_{D10N}$ (small black squares), substrate-free *trans*-A βPGM$_{D10N,P146A}$ (small black diamonds), the *cis*-P βPGM$_{D10N}$:F16BP complex (large gold circles, BMRB 51985), the *trans*-A βPGM$_{D10N,P146A}$:F16BP:MgT complex (large gold diamonds, BMRB 51986), the *trans*-A βPGM$_{D10N,P146A}$:F16BP complex (small gold diamonds, BMRB 51987), the *cis*-P βPGM$_{D10N}$:βG16BP complex (large pink circles, BMRB 27174[29]), the *cis*-P Mg$_{cat}$-free βPGM$_{D10N}$:βG16BP complex (small pink circles, BMRB 27175[29]), the *trans*-A βPGM$_{D10N,P146A}$:βG16BP:MgT complex (large pink diamonds, BMRB 51988), the *cis*-A βPGM$_{D10N,P146A}$:βG16BP complex (small purple circles, BMRB 51989), the *trans*-A βPGM$_{D10N,P146A}$:βG16BP complex (small pink diamonds, BMRB 51990), the *cis*-P βPGM$_{WT}$:MgF$_3$:G6P complex (large blue circles, BMRB 7234[26]) and the *cis*-A βPGM$_{P146A}$:MgF$_3$:G6P complex (small blue circles, BMRB 28097[13]).

binding βG16BP, adoption of a NAC III conformation occurs, which leads to catalysis of proline isomerisation, likely through the stabilisation of a transition state involving a twisted *anti*/C4-*endo* proline ring, and the enzyme population shifts to fully active *cis*-P βPGM.

## Results

### The *cis*-P βPGM$_{D10N}$:F16BP complex adopts a NAC I conformation

The ability of βPGM$_{D10N}$ to produce stable bis-phosphosugar complexes was leveraged to compare the structural consequences of partial allomorphic activation by F16BP and full allomorphic activation by βG16BP. A highly stable *cis*-P βPGM$_{D10N}$:βG16BP complex is purified readily from expression cultures[29]. On removal of βG16BP, substrate-free βPGM$_{D10N}$ in standard NMR buffer is present in solution as a mixed population of *cis*-P βPGM$_{D10N}$ (58%) and *trans*-P βPGM$_{D10N}$ (42%) species that both adopt an open conformation (Fig. 3). This behaviour mirrors that of substrate-free wild-type βPGM under the same conditions (*cis*-P βPGM$_{WT}$ (66%), BMRB 28095[13] and *trans*-P βPGM$_{WT}$ (34%), BMRB 28096[13]). The relative populations of species present simultaneously in solution were calculated using $^1$H$^{15}$N-TROSY peak intensities derived from a substantial number of residues. On addition of F16BP to substrate-free βPGM$_{D10N}$, a *cis*-P βPGM$_{D10N}$:F16BP complex was crystallised and its structure was determined (1.75 Å resolution, PDB 8Q1D) (Fig. 4a, Table 1, Supplementary Fig. 2c). F16BP is bound in a single orientation as the β-anomer, with the axial 2-hydroxyl group and equatorial 3-hydroxyl group hydrogen bonded by the carbonyl group of V47 located in the substrate specificity loop (V36–L53) (Fig. 5a). The K145-P146 peptide bond within the allomorphic control loop (E140–I150) is completely in the *cis* conformation and the alkylammonium sidechain of K145 coordinates the 6-phosphate group of F16BP, which is aligned and in van der Waals contact with the carboxylate nucleophile of D8. The 1-phosphate group is coordinated by R49 in the distal region of the active site. The arrangement of groups in the proximal site in the vicinity of N10 and the catalytic magnesium ion (Mg$_{cat}$) is broadly similar to that observed in the *cis*-P βPGM$_{D10N}$:βG16BP complex (PDB 5OK1[29]) (Fig. 4b, Fig. 5c). Intriguingly, in the *cis*-P βPGM$_{D10N}$:F16BP complex, the sidechain of D170 is rotated by ~15° in a planar manner so that the alternate carboxylate oxygen atom coordinates Mg$_{cat}$. The resulting proximity between the other carboxylate oxygen atom of D170 and the carboxylate sidechain of D8 is close enough to predict protonation of one of

these oxygen atoms. This observation shows that βPGM may be capable of accommodating a proton within a hydrogen bonding network close to the general acid-base residue (Supplementary Fig. 3, Supplementary Note 1).

Notably, the interdomain hinge closure angle observed in the *cis*-P βPGM$_{D10N}$:F16BP complex corresponds better with the domain arrangement of a NAC I conformation (*cis*-P βPGM$_{WT}$:BeF$_3$:G6P complex, PDB 2WF9[27], non-H atom RMSD = 0.9 Å), rather than that of the more closed NAC III conformation (*cis*-P βPGM$_{D10N}$:βG16BP complex, PDB 5OK1[29], non-H atom RMSD = 1.4 Å) (Fig. 1c, d, Supplementary Table 1). This categorisation is confirmed by the distribution of the calculated intrinsic Euler angles, which provide a systematic description of the cap and core interdomain relationship (Fig. 6, Supplementary Data 1). The transition from a NAC I conformation to a NAC III conformation appears to be obstructed by steric clashes between the carbonyl group of V47 and both the axial 2-hydroxyl group and equatorial 3-hydroxyl group of F16BP (Fig. 5b). Altogether, these observations illustrate how the substrate specificity loop is

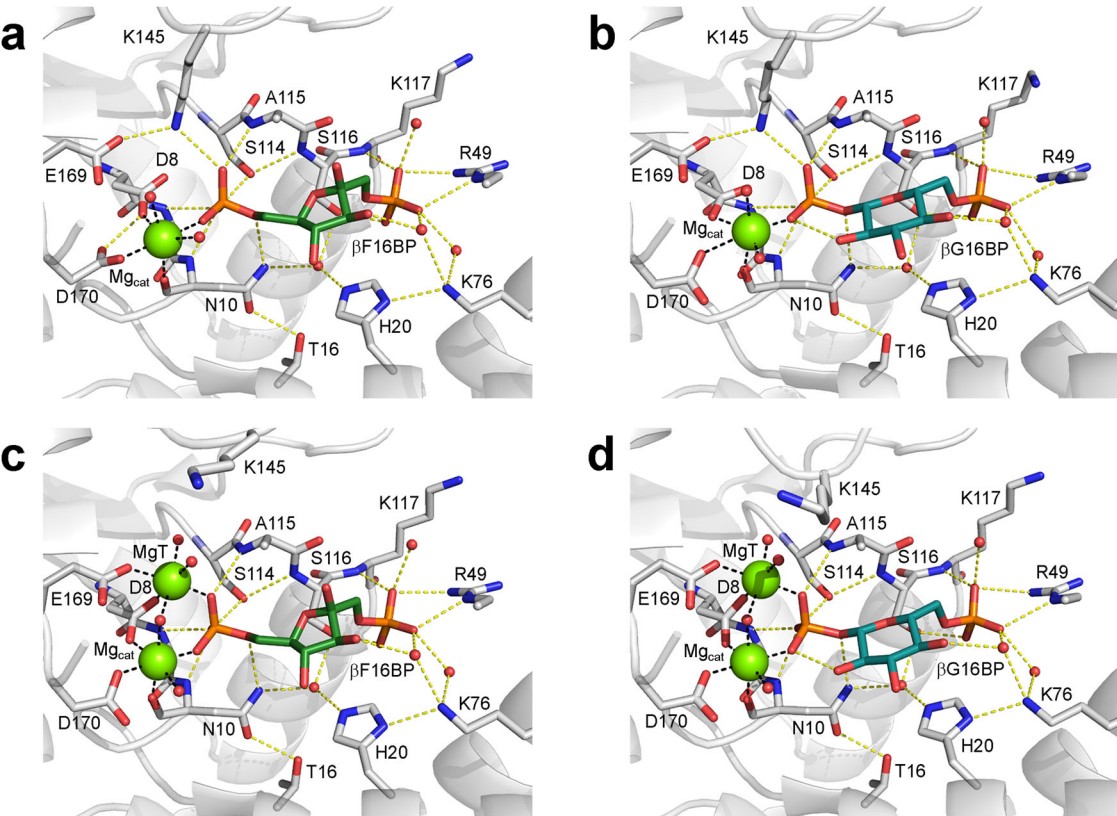

**Fig. 4 | Conformational variation in the active site of βPGM.** Active site details of (**a**) the *cis*-P βPGM$_{D10N}$:F16BP complex (PDB 8Q1D), (**b**) the *cis*-P βPGM$_{D10N}$:βG16BP complex (PDB 5OK1[29]), (**c**) the *trans*-A βPGM$_{D10N,P146A}$:F16BP:MgT complex (PDB 8Q1E) and (**d**) the *trans*-A βPGM$_{D10N,P146A}$:βG16BP:MgT complex (PDB 8Q1F chain A). Selected residues (sticks), together with F16BP (dark green carbon atoms), βG16BP (teal carbon atoms), structural waters (red spheres), Mg$_{cat}$ (green sphere) and MgT (green sphere) are illustrated. Yellow dashes indicate hydrogen bonds and black dashes show metal ion coordination. Apart from R49, residues of the substrate specificity loop (V36–L53) have been omitted for clarity. For the complexes containing a *cis* K145-P146 peptide bond (**a**, **b**), the alkylammonium sidechain of K145 is engaged in the active site, whereas for the complexes containing a *trans* K145-A146 peptide (**c**, **d**), the alkylammonium sidechain occupies a solvent exposed position, with MgT acting as a weakly-binding surrogate for the missing cation.

able to impede the adoption of the NAC III conformation, thereby retarding efficient phosphoryl transfer from this partial allomorphic activator.

### The βPGM$_{D10N}$:F16BP complex populates two forms in solution

The solution properties of the *cis*-P βPGM$_{D10N}$:F16BP complex were investigated to establish the relationship between solution and solid-state behaviour. Addition of 100-fold excess F16BP to substrate-free βPGM$_{D10N}$ in standard NMR buffer (supplemented with 50 mM MgCl$_2$) immediately produced two discrete βPGM$_{D10N}$:F16BP species in slow exchange. The dominant population (86%) was assigned as the *cis*-P βPGM$_{D10N}$:F16BP complex (BMRB 51985), in accord with the crystallisation experiments. The presence of a *cis* K145-P146 peptide bond was confirmed by the $^{13}$Cβ chemical shift of P146 ($^{13}$Cβ = 35.0 ppm[13]) and both A143 and D180 were found to be sensitive reporters of the isomerisation state of this peptide bond (Fig. 3a, b). Weighted chemical shift perturbations (Δδ values) between residues of the *cis*-P βPGM$_{D10N}$:F16BP complex and the *cis*-P βPGM$_{D10N}$:βG16BP complex show large Δδ values in the substrate specificity loop, reflecting the occupancy of F16BP rather than βG16BP in the active site. In addition, large Δδ values in the general acid-base hinge and the 80 s hinge, indicate a difference in the interdomain hinge closure angle between the complexes (Figs. 2, 7a–d). Within the 80 s hinge, both I84 and S88 were found to be sensitive reporters of the interdomain hinge closure angle (in place of D15 used previously[27]), which confirms that the *cis*-P βPGM$_{D10N}$:F16BP complex adopts a NAC I conformation (Fig. 3c, d, Supplementary Fig. 4a), as observed in the crystal (PDB 8Q1D).

While the *cis*-P βPGM$_{D10N}$:F16BP complex forms the dominant population in solution, a second βPGM$_{D10N}$:F16BP species (14%) was observed for which 77 residues were uniquely assignable. Given that substrate-free βPGM$_{D10N}$ exists as a mixed population of *cis*-P βPGM$_{D10N}$ and *trans*-P βPGM$_{D10N}$ species (Fig. 3a, b), a plausible model is that the second βPGM$_{D10N}$:F16BP species is a *trans*-P βPGM$_{D10N}$:F16BP complex. The population of both a *cis*-P βPGM$_{D10N}$:F16BP complex and a *trans*-P βPGM$_{D10N}$:F16BP complex would be consistent with the pronounced lag phase observed in kinetic experiments. To test this hypothesis, a variant was generated containing both a D10N and a P146A substitution (βPGM$_{D10N,P146A}$), in which a *trans* K145-A146 peptide bond should dominate the conformational ensemble[13,34]. P146A variants have low catalytic activity, which also helps alleviate the limited lifetime (~12 h) observed for both βPGM$_{D10N}$:F16BP species, since they convert to the *cis*-P βPGM$_{D10N}$:βG16BP complex on this timescale.

### Substrate-free *trans*-A βPGM$_{D10N,P146A}$ has an open conformation

Using both solution and crystallography techniques, the structural properties of substrate-free βPGM$_{D10N,P146A}$ were investigated. $^1$H$^{15}$N-TROSY spectra of substrate-free βPGM$_{D10N,P146A}$ acquired in standard NMR buffer indicate that it adopts an open conformation and is present in solution as a single *trans*-A βPGM$_{D10N,P146A}$ population, equivalent to substrate-free *trans*-A βPGM$_{P146A}$ (Fig. 3) (BMRB 27920[34]). Substrate-free *trans*-A βPGM$_{D10N,P146A}$ was crystallised (1.7 Å resolution, PDB 8Q1C) (Table 1, Supplementary Fig. 2a, b) and the two monomers in the asymmetric unit both display an open conformation, each of which overlays closely with substrate-free *trans*-A βPGM$_{P146A}$ (PDB 6YDK[13]) (chain A: non-H atom RMSD = 1.5 Å, chain B: non-H atom RMSD = 1.2 Å). In both monomers,

## Table 1 | Data collection and refinement statistics

| | *trans*-A βPGM$_{D10N,P146A}$ | *cis*-P βPGM$_{D10N}$:F16BP | *trans*-A βPGM$_{D10N,P146A}$:F16BP:MgT | *trans*-A βPGM$_{D10N,P146A}$:βG16BP:MgT |
|---|---|---|---|---|
| **Data collection** | | | | |
| Space group | *P*12$_1$1 | *P*2$_1$2$_1$2$_1$ | *P*12$_1$1 | *P*12$_1$1 |
| Cell dimensions | | | | |
| *a, b, c* (Å) | 38.7, 117.3, 53.3 | 31.9, 71.4, 83.9 | 32.1, 83.8, 38.9 | 32.0, 79.7, 79.9 |
| α, β, γ (°) | 90.0, 98.6, 90.0 | 90.0, 90.0, 90.0 | 90.0, 110.7, 90.0 | 90.0, 97.6, 90.0 |
| Resolution (Å) | 117.25 – 1.68 (1.71 – 1.68) * | 54.37 – 1.75 (1.78 – 1.75) | 83.76 – 1.23 (1.25 – 1.23) | *39.83 – 1.20 (1.22 – 1.20)* [a] 39.83 – 1.01 (1.03 – 1.01) |
| $R_{merge}$ | 0.089 (2.317) | 0.151 (1.330) | 0.074 (2.147) | *0.037 (0.198)* [a] 0.038 (0.494) |
| *I* / σ*I* | 13.8 (0.6) | 10.5 (1.1) | 12.3 (0.4) | *31.7 (6.1)* [a] 26.0 (0.8) |
| Completeness (%) | 98.3 (86.9) | 100.0 (99.2) | 100.0 (98.0) | *98.6 (96.3)* [a] 80.3 (11.2) |
| Redundancy | 6.5 (5.7) | 13.1 (13.0) | 6.8 (5.9) | *6.9 (6.9)* [a] 6.3 (1.6) |
| **Refinement** | | | | |
| Resolution (Å) | 1.68 | 1.75 | 1.23 | 1.01 |
| No. reflections | 338981 (12926) | 263042 (13199) | 380047 (15989) | *842130 (41036)* [a] 1046939 (1833) |
| $R_{work}$ / $R_{free}$ | 0.211 / 0.270 | 0.161 / 0.252 | 0.153 / 0.195 | 0.116 / 0.136 |
| No. atoms | | | | |
| Protein | 3374 | 1719 | 1717 | 3527 |
| Ligand / ion | 30 / 2 | 32 / 1 | 24 / 3 | 68 / 7 |
| Water | 185 | 188 | 127 | 485 |
| *B*-factors | | | | |
| Protein—main chain —sidechains | 38 48 | 25 38 | 21 31 | 11 16 |
| Ligand / ion | 40 / 35 | 39 / 53 | 23 / 27 | 16 / 16 |
| Water | 38 | 35 | 30 | 23 |
| R.m.s. deviations | | | | |
| Bond lengths (Å) | 0.0103 | 0.0092 | 0.0082 | 0.0126 |
| Bond angles (°) | 1.671 | 1.567 | 1.524 | 1.757 |

*Values in parentheses are for highest-resolution shell.

[a]The detector was set to 1.2 Å and complete data were collected at this resolution (italic text). However, since high quality incomplete data were also collected to 1.01 Å, all of the data were included in the refinement.

Mg$_{cat}$ is coordinated similarly to substrate-free *cis*-P βPGM$_{WT}$ (PDB 1ZOL[21] and PDB 2WHE[26]) and the position of N10 is analogous to that observed for D10 in substrate-free *trans*-A βPGM$_{P146A}$ (PDB 6YDK[13]). In each active site, an inorganic phosphate anion is coordinated by the side-chains of K117 and R49, along with a tris molecule (derived from the crystallisation buffer) that occupies a similar location to the sugar ring of the allomorphic activators (Supplementary Fig. 2a, b). As expected, both monomers exhibit a *trans* K145-A146 peptide bond with the alkylammonium sidechain of K145 located in a solvent exposed position between the cap and core domains. However, the backbone arrangement of the allomorphic control loop in each monomer is different and neither chain adopts the conformation present in substrate-free *trans*-A βPGM$_{P146A}$ (PDB 6YDK[13]). Such observations imply that the *trans* K145-A146 peptide bond allows access to a broad conformational ensemble for the allomorphic control loop.

### The *trans*-A βPGM$_{D10N,P146A}$:F16BP complex adopts a NAC I conformation

Substrate-free βPGM$_{D10N,P146A}$ was crystallised in the presence of F16BP to investigate its binding properties and a structure of the resulting *trans*-A βPGM$_{D10N,P146A}$:F16BP:MgT complex was determined (1.2 Å resolution, PDB 8Q1E) (Figs. 4c, 5d, Table 1, Supplementary Fig. 2d). The β-anomer of F16BP is coordinated in the active site in a similar arrangement to that observed in the *cis*-P βPGM$_{D10N}$:F16BP complex (PDB 8Q1D). In contrast,

the sidechain of K145 is in a solvent exposed position and a water molecule now occupies the vacated alkylammonium cation site. An additional magnesium ion (MgT) is coordinated between this water molecule and Mg$_{cat}$, providing charge complementarity in the neighbourhood of the 6-phosphate group of F16BP. The *trans*-A βPGM$_{D10N,P146A}$:F16BP:MgT complex (PDB 8Q1E) overlays closely with the *cis*-P βPGM$_{D10N}$:F16BP complex (PDB 8Q1D) (non-H atom RMSD = 1.2 Å) and the interdomain hinge closure angle and Euler angles each correspond to a NAC I conformation (Fig. 6, Supplementary Table 1). Again, further closure towards a NAC III conformation appears to be obstructed by steric clashes between the carbonyl group of V47 and both the axial 2-hydroxyl group and equatorial 3-hydroxyl group of F16BP (Fig. 5e).

### The *trans*-A βPGM$_{D10N,P146A}$:F16BP complex populates two forms in solution

Addition of 100-fold excess F16BP to substrate-free *trans*-A βPGM$_{D10N,P146A}$ in standard NMR buffer immediately produced two discrete βPGM$_{D10N,P146A}$:F16BP species in slow exchange. The dominant population (72%) was assigned as the *trans*-A βPGM$_{D10N,P146A}$:F16BP:MgT complex (BMRB 51986) (Fig. 3a, b, Supplementary Fig. 4b), in accord with the crystallisation experiments. The presence of MgT was corroborated by the behaviour of the backbone amide proton of A115, which resonates upfield by 1.6 ppm compared to its position in the *cis*-P βPGM$_{D10N}$:βG16BP complex (BMRB 27174[29]) (Supplementary Data 2, Supplementary Data 3).

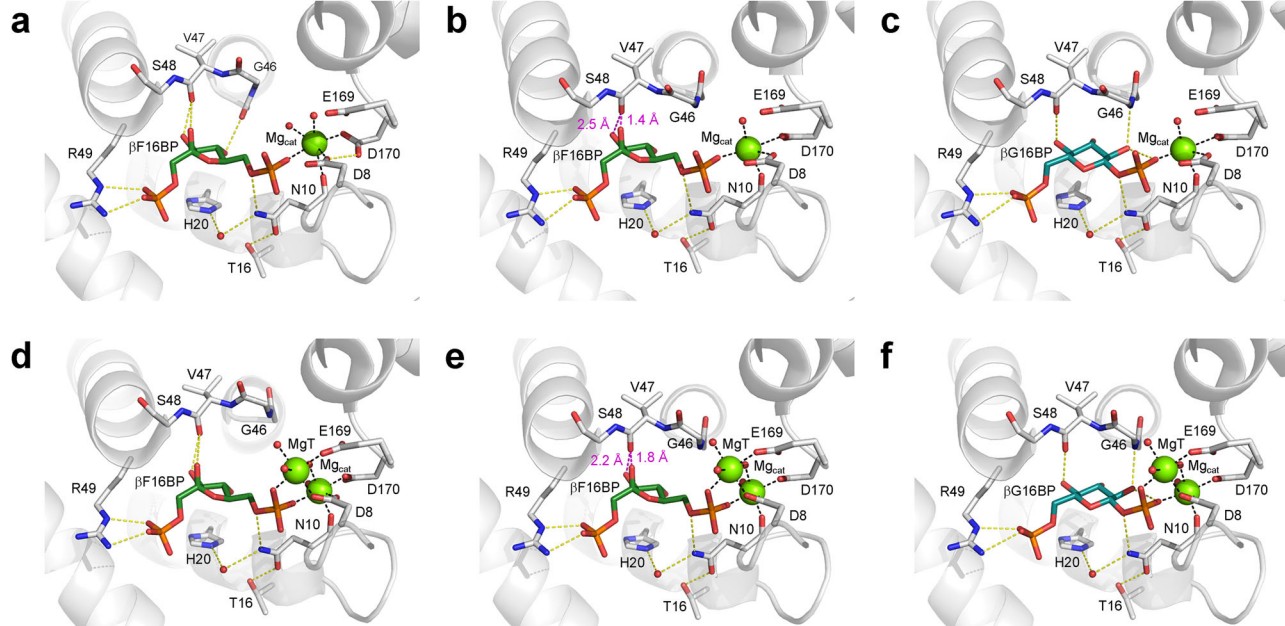

**Fig. 5 | A steric clash is introduced between F16BP and V47 on the transition from a NAC I conformation to a NAC III conformation.** Active site details of (**a**) the NAC I conformation of the *cis*-P βPGM$_{D10N}$:F16BP complex (PDB 8Q1D), (**b**) a model with F16BP replacing βG16BP within the NAC III conformation of the *cis*-P βPGM$_{D10N}$:βG16BP complex (PDB 5OK1[29]), (**c**) the NAC III conformation of the *cis*-P βPGM$_{D10N}$:βG16BP complex (PDB 5OK1[29]), (**d**) the NAC I conformation of the *trans*-A βPGM$_{D10N,P146A}$:F16BP:MgT complex (PDB 8Q1E), (**e**) a model with F16BP replacing βG16BP within the NAC III conformation of the *trans*-A βPGM$_{D10N,P146A}$:βG16BP:MgT complex (PDB 8Q1F chain B) and (**f**) the NAC III conformation of the *trans*-A βPGM$_{D10N,P146A}$:βG16BP:MgT complex (PDB 8Q1F chain B). Selected residues (sticks), together with F16BP (dark green carbon atoms),

βG16BP (teal carbon atoms), structural waters (red spheres), Mg$_{cat}$ (green sphere) and MgT (green sphere) are illustrated. Yellow dashes indicate hydrogen bonds and black dashes show metal ion coordination. Magenta dashes and labels (**b, e**) specify heavy atom distances that comprise steric clashes between the carbonyl group of V47 and both the axial 2-hydroxyl group and the equatorial 3-hydroxyl group of F16BP on adoption of a NAC III conformation. Such close proximity between F16BP and residues of the substrate specificity loop (V36–L53) impedes the transition from a NAC I conformation to a NAC III conformation. G46 is highly conserved across members of the haloacid dehalogenase superfamily[33], where it serves to coordinate βG16BP in a NAC III conformation (**c, f**) but has positional variability in a NAC I conformation (**a, d**).

The second species was assigned as a *trans*-A βPGM$_{D10N,P146A}$:F16BP complex (BMRB 51987) without MgT. Only small Δδ values are observed between the two complexes, which are localised to the substrate specificity loop, the allomorphic control loop, Mg$_{cat}$ site and 170 s α-helix, though the backbone amide proton of A115 is not observable (Fig. 2, Supplementary Fig. 5a–d). From the populations measured and the concentration of Mg$^{2+}$ (5 mM), the MgT dissociation constant was estimated as $K_d$ (MgT) ~ 2 mM. Both complexes adopt a NAC I conformation (Fig. 3c, d, Supplementary Fig. 4a).

The solution properties of the *trans*-A βPGM$_{D10N,P146A}$:F16BP:MgT complex and the *cis*-P βPGM$_{D10N}$:F16BP complex corroborate the conformational features identified in their respective crystal structures (PDB 8Q1E, PDB 8Q1D, Figs. 4a, c, 5a, d, Supplementary Fig. 2c, d). Large Δδ values are only observed for the allomorphic control loop confirming that the principal difference between these complexes is the isomerisation state of the K145-X146 peptide bond (Figs. 2, 7a, b, e, f). Moderate Δδ values are observed for the Mg$_{cat}$ site, the 170 s α-helix and the substrate specificity loop, in line with the withdrawal of the alkylammonium sidechain of K145 from the active site. In contrast, negligible Δδ values are present for residues comprising the 80 s hinge, indicating that the interdomain hinge closure angle is consistent between the complexes and represent NAC I conformations (Fig. 3c, d, Supplementary Fig. 4a). Overall, these results indicate that NAC I conformations are preferred for F16BP complexes irrespective of the isomerisation state of the K145-X146 peptide bond.

### Population of a *trans*-P βPGM$_{D10N}$:F16BP:MgT complex supports the mechanism of partial allomorphic activation

Confirmation of the identity of the second βPGM$_{D10N}$:F16BP species (14%) that was predicted to be a *trans*-P βPGM$_{D10N}$:F16BP complex was achieved using the chemical shift assignments of the *trans*-A

βPGM$_{D10N,P146A}$:F16BP:MgT complex (BMRB 51986), the *trans*-A βPGM$_{D10N,P146A}$:F16BP complex (BMRB 51987) and the *cis*-P βPGM$_{D10N}$:F16BP complex (BMRB 51985) (Supplementary Fig. 5a, b, e–j, Supplementary Fig. 6, Supplementary Data 4). Pearson correlation analysis using both the backbone amide proton and amide nitrogen chemical shifts of the 77 residues that could be uniquely assigned revealed a near-perfect correlation with the *trans*-A βPGM$_{D10N,P146A}$:F16BP:MgT complex ($r$ = 0.9998 for the amide proton comparison and $r$ = 0.9999 for the amide nitrogen comparison). The binding of MgT was further corroborated by the behaviour of the backbone amide proton of A115, which resonates upfield by 1.6 ppm compared to its position in the *cis*-P βPGM$_{D10N}$:βG16BP complex (BMRB 27174[29]). This analysis confirms the identity of the species as a *trans*-P βPGM$_{D10N}$:F16BP:MgT complex with solution properties that are fully consistent with a *trans* K145-P146 peptide bond, and the adoption of a NAC I conformation. Hence, the substantial population of both a *cis*-P βPGM$_{D10N}$:F16BP complex and a *trans*-P βPGM$_{D10N}$:F16BP:MgT complex provides strong evidence that F16BP binding does not strongly affect the ratio of *cis*-P to *trans*-P forms meaning that F16BP is only a partial allomorphic activator.

### The βPGM$_{D10N,P146A}$:βG16BP complex adopts a NAC III conformation

The structural processes underpinning accelerated isomerisation of the K145-P146 peptide bond on full allomorphic activation by βG16BP were investigated further. Substrate-free βPGM$_{D10N,P146A}$ was crystallised in the presence of βG16BP and a structure of the resulting *trans*-A βPGM$_{D10N,P146A}$:βG16BP:MgT complex was determined (1.0 Å resolution, PDB 8Q1F) (Figs. 4d, 5f, Table 1, Supplementary Fig. 2e, f). Two monomers are present in the asymmetric unit, each of which overlays closely with the *cis*-P βPGM$_{D10N}$:βG16BP complex (PDB 5OK1[29]) (chain A: non-H atom

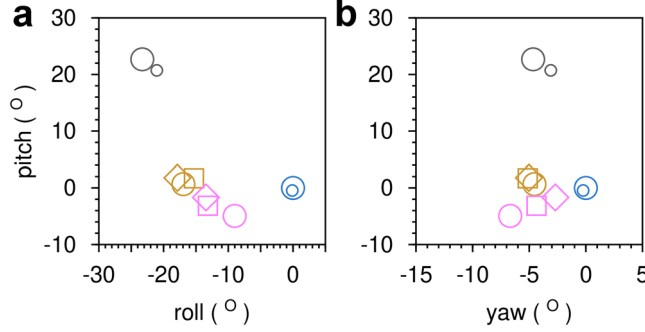

**Fig. 6 | Intrinsic Euler angles describing the cap and core interdomain relationship for selected βPGM crystal structures with respect to the *cis*-P βPGM$_{WT}$:MgF$_3$:G6P complex. a** Roll angle (cap and core twisting motion) and pitch angle (cap and core closing angle). **b** Yaw angle (cap and core left-to-right lateral rotation) and pitch angle (cap and core closing angle). The βPGM species compared are: Open conformation: substrate-free *cis*-P βPGM$_{WT}$ (large black circles, PDB 2WHE[26], small black circles, PDB 1ZOL[21]). NAC I conformation: *cis*-P βPGM$_{WT}$:BeF$_3$:G6P complex (gold circles, PDB 2WF9[27]), *cis*-P βPGM$_{D10N}$:F16BP complex (gold squares, PDB 8Q1D) and *trans*-A βPGM$_{D10N,P146A}$:F16BP:MgT complex (gold diamonds, PDB 8Q1E). NAC III conformation: *cis*-P βPGM$_{D10N}$:βG16BP complex (pink circles, PDB 5OK1[29]), *trans*-A βPGM$_{D10N,P146A}$:βG16BP:MgT complex (pink squares, PDB 8Q1F chain A) and *trans*-A βPGM$_{D10N,P146A}$:βG16BP:MgT complex (pink diamonds, PDB 8Q1F chain B). Both chains of the *trans*-A βPGM$_{D10N,P146A}$:βG16BP:MgT complex crystal structure are denoted as NAC III$^t$ conformations. Fully closed near-transition state conformation: *cis*-P βPGM$_{WT}$:MgF$_3$:G6P complex (large blue circles, PDB 2WF5[26]) and *cis*-A βPGM$_{P146A}$:MgF$_3$:G6P complex (small blue circles. PDB 6YDJ[13]).

RMSD = 1.3 Å, chain B: non-H atom RMSD = 1.3 Å) and their corresponding interdomain hinge closure angle and Euler angles are consistent with a NAC III conformation. Chain A is slightly more closed than chain B (Fig. 6, Supplementary Table 1). In each active site, βG16BP is coordinated by the substrate specificity loop with the 1-phosphate group aligned and in van der Waals contact with the carboxylate nucleophile of D8, while the 6-phosphate group is coordinated by R49 in the distal region of the active site. The K145-A146 peptide bond is in the *trans* conformation and the sidechain of K145 is in a solvent exposed position. A water molecule occupies the vacated alkylammonium cation site, and MgT is coordinated between this water molecule and Mg$_{cat}$. The observation of MgT in the same site within the *trans*-A βPGM$_{D10N,P146A}$:βG16BP:MgT complex, the *trans*-P βPGM$_{D10N}$:F16BP:MgT complex and the *trans*-A βPGM$_{D10N,P146A}$:F16BP:MgT complex indicates that its accommodation is correlated with a *trans* K145-X146 peptide bond. However, it is independent of the identity of the bound bis-phosphosugar or the adoption of either a NAC I or NAC III conformation. Additionally, a correspondence in bond alignment and atom position between the *trans* K145-A146 peptide bond of the *trans*-A βPGM$_{D10N,P146A}$:βG16BP:MgT complex (PDB 8Q1F) and the *cis* K145-P146 peptide bond of the *cis*-P βPGM$_{D10N}$:βG16BP complex (PDB 5OK1[29]) allows a model of the *trans* K145-P146 peptide bond to be proposed with confidence (Supplementary Fig. 7).

In each monomer of the *trans*-A βPGM$_{D10N,P146A}$:βG16BP:MgT complex (PDB 8Q1F), different backbone conformations are observed for residues V141–K145 of the allomorphic control loop. Chain A mirrors the conformation observed in the *trans*-A βPGM$_{D10N,P146A}$:F16BP:MgT complex (PDB 8Q1E), whereas chain B has a close correspondence to substrate-free *trans*-A βPGM$_{P146A}$ (PDB 6YDK[13]). For the allomorphic control loop of chain A, the difference electron density map shows that these residues likely also adopt the conformation seen in chain B, but with low occupancy. The remainder of the allomorphic control loop (residues A146–I150) and the specific geometry of the K145-A146 dipeptide segment is consistent between chains. However, despite the slightly more open interdomain hinge closure angle in chain B (Supplementary Table 1), the conformation of the allomorphic control loop allows the formation of a hitherto unobserved

interdomain hydrogen bond with the substrate specificity loop (K145$_{NH}$–G46$_{CO}$) (Fig. 8a, c). These loops are too distant for direct interdomain hydrogen bonding in chain A.

## The βPGM$_{D10N,P146A}$:βG16BP complex populates multiple forms in solution

Following the addition of tenfold excess βG16BP to substrate-free *trans*-A βPGM$_{D10N,P146A}$ in standard NMR buffer, three distinct long-lived species are observed in slow exchange. The relative populations of these species exhibited a Mg$^{2+}$-dependency, which was exploited to assist NMR resonance assignment. The isomerisation state of the K145-A146 peptide bond in each complex was identified using the chemical shift behaviour of A143 and D180 (Fig. 3a, b), and the presence of MgT was confirmed by the backbone amide proton of A115 (Supplementary Data 2, Supplementary Data 3). The three species were identified as the *trans*-A βPGM$_{D10N,P146A}$:βG16BP:MgT complex (BMRB 51988), in accord with the crystallisation experiments, a *trans*-A βPGM$_{D10N,P146A}$:βG16BP complex without MgT (BMRB 51990) and, unexpectedly, a *cis*-A βPGM$_{D10N,P146A}$:βG16BP complex (BMRB 51989). The populations in standard NMR buffer were 44%, 16%, and 40%, respectively. Hence, the binding of βG16BP to substrate-free *trans*-A βPGM$_{D10N,P146A}$ is able to stabilise a species with a *cis* K145-A146 peptide bond (Supplementary Fig. 4b).

The properties of the *cis*-A βPGM$_{D10N,P146A}$:βG16BP complex (BMRB 51989) were investigated through comparison with the *cis*-P βPGM$_{D10N}$:βG16BP complex (Fig. 7a, b, g, h) (BMRB 27174[29]). Almost negligible Δδ values are observed for all residues, apart from around residue 146, which indicates that both complexes adopt indistinguishable NAC III conformations (Fig. 3c, d, Supplementary Fig. 4a), with Mg$_{cat}$ bound equivalently in each active site. Moreover, all residues of the *cis*-A βPGM$_{D10N,P146A}$:βG16BP complex were assignable in the $^1$H$^{15}$N-TROSY spectrum, indicating that millisecond exchange is not present to a large degree, which points to an inherent stability within this conformer (Supplementary Fig. 8, Supplementary Note 2).

Comparison between the *cis*-A βPGM$_{D10N,P146A}$:βG16BP complex (BMRB 51989) and the *trans*-A βPGM$_{D10N,P146A}$:βG16BP complex (BMRB 51990) indicates that although each complex adopts a NAC III conformation (Fig. 3d, Supplementary Fig. 4a), small Δδ values for the 80 s hinge show that the complexes populate definably different conformations (Fig. 3c, Supplementary Fig. 9a–d). Furthermore, the equivalent comparison between the *trans*-A βPGM$_{D10N,P146A}$:βG16BP:MgT complex (BMRB 51988) and the *trans*-A βPGM$_{D10N,P146A}$:βG16BP complex (BMRB 51990) reveals that both complexes adopt the same NAC III conformation. Hence, this new conformation is a feature of the *trans* K145-A146 peptide bond and is independent of the presence of MgT (Fig. 3c, Supplementary Fig. 9a, b, e, f). These different NAC III conformations populated in solution are also reflected by the crystal structures of the *cis*-P βPGM$_{D10N}$:βG16BP complex (PDB 5OK1[29]) and the *trans*-A βPGM$_{D10N,P146A}$:βG16BP:MgT complex (PDB 8Q1F). Within the 80 s hinge, the length of the I84$_{NH}$–Y80$_{CO}$ hydrogen bond is measurably different between the complexes, mirroring the chemical shift behaviour of I84 (Fig. 3c). This observation correlates with a twist of the cap domain relative to the core domain, while maintaining Euler angles within the range of NAC III conformations (Fig. 6). Hence, the new NAC III conformer adopted by the two *trans*-A βPGM$_{D10N,P146A}$:βG16BP complexes is denoted here as a NAC III$^t$ conformation.

Further comparison between the NAC III and NAC III$^t$ conformations reveals Δδ values propagated to residues of the general acid-base hinge, the substrate specificity loop, the phosphodianions bridging loop, the allomorphic control loop and in the vicinity of the Mg$_{cat}$ site (Fig. 2, Supplementary Fig. 9a–h). Notably, complexes with a *cis* K145-P146 peptide bond are also able to populate a NAC III$^t$ conformation. Comparison of the *cis*-P Mg$_{cat}$-free βPGM$_{D10N}$:βG16BP complex (BMRB 27175[29]) with the *cis*-P βPGM$_{D10N}$:βG16BP complex (BMRB 27174[29]), shows similar small Δδ values for the 80 s hinge (Supplementary Fig. 9a, b, i, j). In particular, the chemical shift behaviour of I84 for the *cis*-P Mg$_{cat}$-free βPGM$_{D10N}$:βG16BP

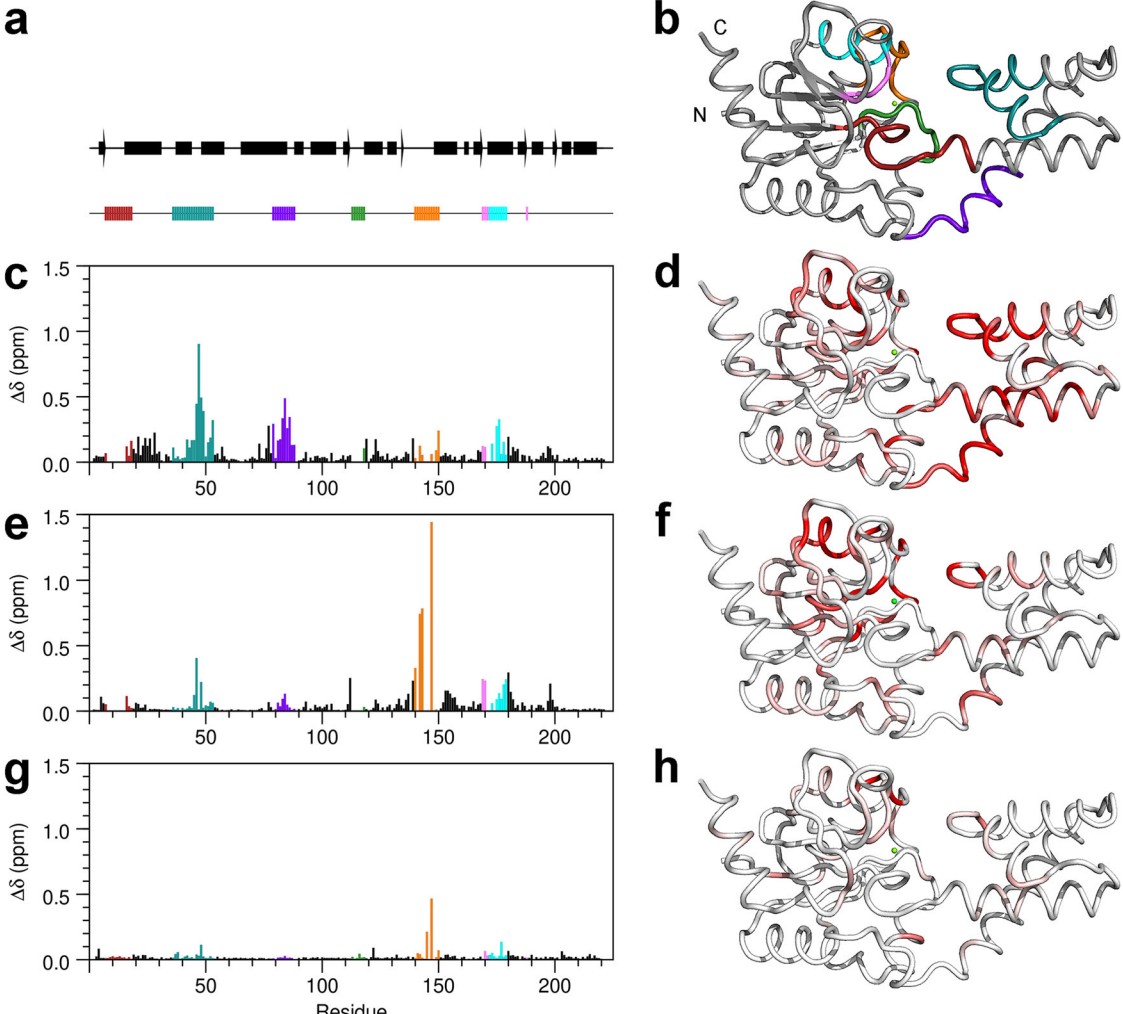

**Fig. 7 | Weighted chemical shift perturbations reporting differences in the solution conformations of βPGM complexes. a, b** Scheme showing the architecture of substrate-free *cis*-P βPGM_WT (PDB 2WHE[26]) with secondary structure elements indicated by bars (α-helices) and arrows (β-strands). Active site regions are highlighted by coloured bars and coloured cartoon backbone: general acid-base hinge (dark red, F7–E18), substrate specificity loop (teal, V36–L53), 80 s hinge (purple, N79–S88), phosphodianions bridging loop (green, A113–N118), allomorphic control loop (orange, E140–I150), Mg_cat site (pink, E169–S171, V188) and 170 s α-helix (cyan, Q172–K179). **c–h** Weighted chemical shift perturbations of the backbone amide group are calculated for each residue as: $\Delta\delta = [(\delta_{HN\text{-}X} - \delta_{HN\text{-}Y})^2 + (0.13 \times (\delta_{N\text{-}X} - \delta_{N\text{-}Y}))^2]^{1/2}$, where X and Y are the two βPGM complexes being compared. $\Delta\delta$ values are shown by coloured histogram bars and red-shaded cartoon backbone. **c, d** $\Delta\delta$ values between the *cis*-P βPGM_D10N:F16BP complex (BMRB 51985) and the *cis*-P

βPGM_D10N:βG16BP complex (BMRB 27174[29]). Large $\Delta\delta$ values in the substrate specificity loop reflect the occupancy of F16BP, rather than βG16BP in the active site and large $\Delta\delta$ values in the general acid-base hinge and the 80 s hinge indicate a difference in the interdomain hinge closure angle between the complexes. **e, f** $\Delta\delta$ values between the *trans*-A βPGM_D10N,P146A:F16BP:MgT complex (BMRB 51986) and the *cis*-P βPGM_D10N:F16BP complex (BMRB 51985). Large $\Delta\delta$ values in the allomorphic control loop indicate that the principal difference between these complexes is the isomerisation state of the K145-X146 peptide bond. Smaller $\Delta\delta$ values arise from the differential occupancy of MgT. **g, h** $\Delta\delta$ values between the *cis*-A βPGM_D10N,P146A:βG16BP complex (BMRB 51989) and the *cis*-P βPGM_D10N:βG16BP complex (BMRB 27174[29]). Almost negligible $\Delta\delta$ values are observed for all residues, apart from those in the immediate vicinity of the site of substitution at residue 146, indicating that these complexes adopt identical NAC III conformations.

complex mirrors that observed for the *trans*-A βPGM_D10N,P146A:βG16BP complex (Fig. 3c). Therefore, these observations point to roles for both the isomerisation state of the K145-X146 peptide bond and Mg_cat in the transition between the NAC III conformers.

## Discussion

Allomorphic control of βPGM involves the isomerisation of the K145-P146 peptide bond situated within an allomorphic control loop, which leads to either fully active *cis*-P βPGM or partially active *trans*-P βPGM. A change in their relative population acts to regulate the overall catalytic rate. Partial allomorphic activation by F16BP results in phosphoryl transfer to both species, producing *cis*-P βPGM^P and *trans*-P βPGM^P, and manifests as a pronounced lag phase in activity until the population of *cis*-P βPGM^P

dominates[13] (Fig. 1a, Supplementary Fig. 1b). The concurrent population of two phosphorylated species indicates that the phosphoryl transfer rate is faster than isomerisation of the K145-P146 peptide bond, which interconverts at a rate between 0.003 s⁻¹ and 1.0 s⁻¹ in substrate-free βPGM. In contrast, full allomorphic activation by βG16BP indicates that the rate of *trans* to *cis* isomerisation of the K145-P146 peptide bond is accelerated more than *ca.* 1000-fold, such that it occurs faster than the phosphoryl transfer rate. Thus, *cis*-P βPGM^P is generated rapidly, which produces a linear initial rate profile[13] (Fig. 1b, Supplementary Fig. 1b).

The remarkable rate enhancement observed for the βG16BP-mediated isomerisation of the allomorphic control loop indicates that the conversion from *trans*-P βPGM to *cis*-P βPGM progresses through a defined catalytic mechanism with its own stabilised transition state. Structures representing

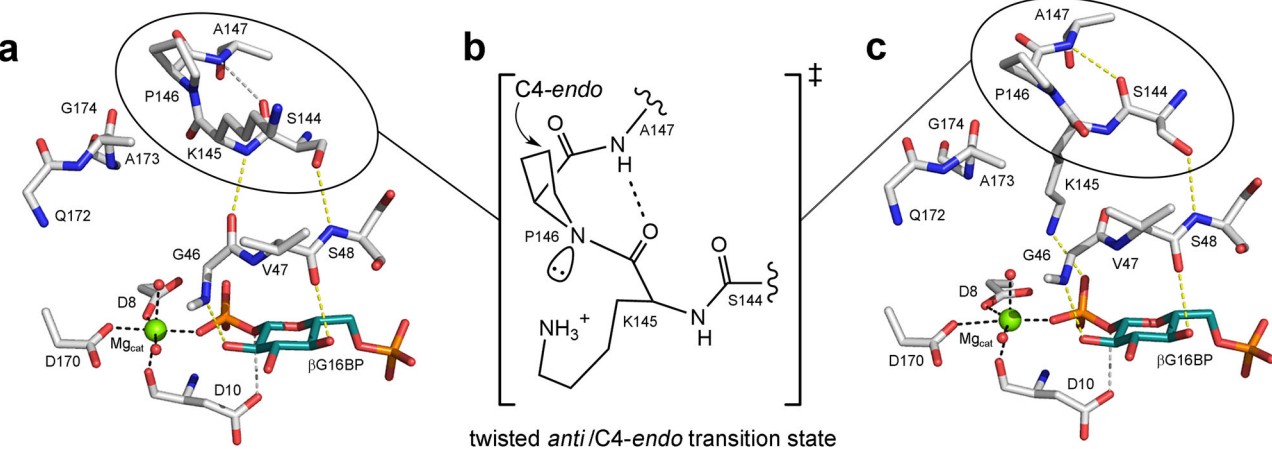

twisted *anti*/C4-*endo* transition state

**Fig. 8 | βG16BP-mediated isomerisation of the K145-P146 peptide bond within the allomorphic control loop passes through a twisted *anti*/C4-*endo* transition state. a** Model of the active site of a *trans*-P βPGM_WT:βG16BP complex (adapted from the *trans*-A βPGM_D10N,P146A:βG16BP:MgT complex, PDB 8Q1F chain B) in a NAC III conformation prior to phosphoryl transfer. There is a close approach between residues of the substrate specificity loop (G46, V47 and S48) and the allomorphic control loop (S144 and K145), which partially rotates the S144-K145 peptide moiety and distorts the A147_NH–S144_CO hydrogen bond across the type VIa β-turn. **b** Proposed model of the transient twisted *anti*/C4-*endo* transition state. Isomerisation of the K145-P146 peptide bond requires a reduction in the partial double bond character through the development of tetrahedral geometry at the P146 nitrogen atom. The transient

nitrogen lone pair forms on the outer face of the β-turn due to the C4-*endo* pucker of the P146 ring and is likely stabilised by the alkylammonium sidechain of K145. Simultaneously, the carbonyl group of K145 can reposition within the confines of the β-turn, becoming hydrogen bonded with the backbone amide group of A147. **c** Model of the active site of a *cis*-P βPGM_WT:βG16BP complex (adapted from the *cis*-P βPGM_D10N:βG16BP complex, PDB 5OK1[29]) in a NAC III conformation prior to phosphoryl transfer. **a**, **c** Selected residues (sticks), together with βG16BP (teal carbon atoms), structural waters (red spheres) and Mg_cat (green sphere) are shown. Yellow dashes indicate hydrogen bonds (≤3.0 Å), grey dashes indicate hydrogen bonds (>3.0 Å) and black dashes show metal ion coordination. Black ovals highlight the region depicted in (**b**).

before and after the isomerisation step are provided by the *trans*-A βPGM_D10N,P146A:βG16BP:MgT complex (PDB 8Q1F, BMRB 51988) and the *cis*-P βPGM_D10N:βG16BP complex (PDB 5OK1, BMRB 27174[29]), respectively. βPGM_D10N complexes form near-attack conformers with βG16BP, in which the nucleophile and the electrophile are aligned and in van der Waals contact, and the balance between the *cis* and *trans* forms of the K145-X146 peptide bond in these conformers is perturbed using the P146A substitution (Figs. 3, 4b, d, 5c, f, 6, Supplementary Fig. 4a, Supplementary Table 1). Importantly, all βG16BP complexes adopt NAC III conformations, implying that isomerisation of the allomorphic control loop is accomplished within this conformation, prior to the structural transformation towards the conformer that supports the transition state for phosphoryl transfer.

Insights into this mechanism come from the behaviour of the two chains in the *trans*-A βPGM_D10N,P146A:βG16BP:MgT complex, compared with the *cis*-P βPGM_D10N:βG16BP complex. In one chain, a small modulation in the interdomain hinge angle leads to a close approach of the substrate specificity loop and the allomorphic control loop, which results in the formation of two direct interdomain hydrogen bonds (Fig. 8a, c). The S144–K145 peptide moiety is also partially rotated, distorting the A147_NH–S144_CO β-turn hydrogen bond. Rapid interconversion between the conformations identified in these crystal structures is supported by millisecond exchange behaviour in solution (Supplementary Fig. 8, Supplementary Note 2).

Generally, peptide bond isomerisation involves a reduction in partial double bond character through the development of tetrahedral geometry at the backbone nitrogen atom, leading to a twisted transition state in which the peptide bond angle is around 90° [35,36]. Rotation of the K145–P146 peptide bond in βPGM_WT towards such a transition state would entail the transient nitrogen lone pair forming on the outer face of the β-turn due to the C4-*endo* pucker of the P146 ring (Fig. 8). Here, the lone pair is in a position where it can interact with the alkylammonium group of K145, which would lead to stabilisation of the transition state[35]. Simultaneously, the carbonyl group of K145 can reposition within the distorted S144–A147 β-turn and become hydrogen bonded with the

backbone amide group of A147. This particular sense of rotation is designated anticlockwise (*anti*) and therefore isomerisation of the K145-P146 peptide bond in βPGM would pass through a twisted *anti*/C4-*endo* transition state.

Such a catalytic mechanism of proline isomerisation contrasts with that associated with, for example, members of the peptidyl prolyl *cis-trans* isomerases (PPIases), which stabilise a twisted *syn*/*exo* transition state[37,38]. However, P146 has a C4-*endo* pucker in all structures of βPGM, and a clockwise rotation (*syn*) would lead to extensive steric clashes. Interestingly, the ability of βG16BP to catalyse the isomerisation of a K145-X146 peptide bond extends beyond proline, at least to alanine, despite the strong conformational preference towards a *trans* K145-A146 peptide bond in substrate-free βPGM_P146A and substrate-free βPGM_D10N,P146A. βPGM_P146A shows a linear initial rate profile in kinetic experiments when βG16BP is used as an allomorphic activator[13], and the observation in solution of a *cis*-A βPGM_D10N,P146A:βG16BP complex (BMRB 51989) both indicate that βG16BP is capable of transforming the K145-A146 peptide bond into the fully active form (Fig. 3a, b, Supplementary Fig. 4b).

In contrast to βG16BP, F16BP readily forms both *trans*-P and *cis*-P (PDB 8Q1D, BMRB 51985) complexes with βPGM_D10N, though the *trans*-P form is more conveniently studied using the *trans*-A βPGM_D10N,P146A:F16BP:MgT complex (PDB 8Q1E, BMRB 51986). Furthermore, unlike βG16BP, all F16BP complexes adopt NAC I conformations rather than NAC III conformations, though alignment and van der Waals contact between nucleophile and electrophile is retained (Figs. 3, 4a, c, 5a, d, 6, Supplementary Fig. 4a, Supplementary Table 1). It appears that progression towards a NAC III conformation is impeded by what would be steric clashes generated between the substrate specificity loop and two hydroxyl groups of F16BP (Fig. 5b, e). Instead, the position of reacting groups along with the ready population of both *cis*-P βPGM and *trans*-P βPGM complexes implies that phosphoryl transfer from F16BP occurs from a NAC I conformation, with a corresponding pronounced retardation in βPGM_WT catalytic activity (Fig. 1a, Supplementary Fig. 1b). In line with this low catalytic activity, βPGM fails to produce a well-defined transition state analogue complex containing a

metallofluoride moiety and F6P or βF1P, using standard NMR or crystallography protocols[26,28,39,40] (Supplementary Fig. 10).

A consistent feature revealed by both βG16BP and F16BP complexes of βPGM variants that populate a *trans* K145-X146 peptide bond is the ability of a magnesium ion (MgT) to bind, albeit weakly, in place of the alkyllammonium group of K145 in the active site (Fig. 4c, d). The binding of MgT can also be inferred in kinetic experiments involving βPGM$_{WT}$, where the initial rate of G6P production is retarded substantially by elevated MgCl$_2$ concentrations[41]. Under such conditions, βG16BP-mediated isomerisation of the K145-P146 peptide bond appears to enable population of the *trans*-P βPGM$_{WT}$:βG16BP:MgT complex as part of the ensemble of enzyme species, i.e. βG16BP accelerates *cis* to *trans* isomerisation thereby exposing a binding site for MgT.

In summary, βG16BP mediates catalysis of proline isomerisation to the *cis*-P form within a NAC III conformation, and delivers full allomorphic activation of βPGM prior to the phosphoryl transfer chemical step. Unusually, catalysis of this isomerisation likely occurs via a twisted *anti*/C4-*endo* transition state. The partial allomorphic activator F16BP fails to stabilise this NAC III conformation and instead arrests βPGM at a NAC I conformation, thereby allowing phosphoryl transfer to both *cis*-P βPGM and *trans*-P βPGM.

## Methods

### Reagents
Unless stated otherwise, reagents were purchased from Merck, GE Healthcare, Melford Laboratories or CortecNet.

### βPGM expression and purification
The βPGM$_{D10N}$ and βPGM$_{D10N,P146A}$ gene sequences were created by site-directed mutagenesis (QuikChange II kit, Agilent Technologies) of the *pgmB* gene (encoding βPGM$_{WT}$) from *Lactococcus lactis* (subspecies *lactis* IL1403) (NCBI: 1114041) cloned within a pET22b+ vector. For βPGM$_{D10N}$, primers with single site base changes encoding the D10N residue substitution[29] were used to modify the βPGM$_{WT}$ gene, whereas for βPGM$_{D10N,P146A}$, primers with single site base changes encoding the P146A residue substitution[34] were used to modify the βPGM$_{D10N}$ gene. Successful mutagenesis was confirmed by DNA sequencing. The βPGM$_{D10N}$ and βPGM$_{D10N,P146A}$ plasmids were transformed into *Escherichia coli* strain BL21(DE3) cells (Novagen) and expressed in defined $^{15}$N or $^2$H$^{15}$N$^{13}$C isotopically enriched M9 minimal media[42] to obtain uniformly $^{15}$N-labelled or $^2$H$^{15}$N$^{13}$C-labelled protein. Cultures were grown at 37 °C with shaking until OD$_{600}$ = 0.6, then cooled at 25 °C and induced with 0.5 mM isopropyl β-D-1-thiogalactopyranoside (IPTG) for a further 18 h. Cells were harvested by centrifugation at 15,000 × *g* for 10 min (Beckman Coulter Avanti centrifuge, Rotor: JA-14). The cell pellet was resuspended in ice-cold standard purification buffer (50 mM K$^+$ HEPES (pH 7.2), 5 mM MgCl$_2$, 2 mM NaN$_3$ and 1 mM EDTA) supplemented with cOmplete™ protease inhibitor cocktail and lysed by 6 × 20 s cycles of sonication (Fisherbrand Model 505 Sonic Dismembrator, 30% amplitude). The cell lysate was cleared by centrifugation at 48,000 × *g* for 35 min at 4 °C (Beckman Coulter Avanti centrifuge, Rotor: JA-20). The soluble fraction was filtered using a 0.22 μm syringe filter and loaded onto a DEAE-Sepharose fast flow anion-exchange column connected to an ÄKTA Prime purification system, which had been washed previously with 1 M NaOH and 6 M guanidine hydrochloride and equilibrated with five column volumes of standard purification buffer. Bound proteins were eluted using a gradient of 0% to 50% standard purification buffer supplemented with 1 M NaCl over 300 mL. Fractions containing βPGM were identified by SDS-PAGE and concentrated to a 5–10 mL volume with a Vivapin (10 kDa molecular weight cut off, Sartorius) using a benchtop centrifuge operating at 3400 × *g* and 4 °C (Thermo Scientific Heraeus Labofuge 400 R). The concentrated protein sample was loaded onto a prepacked Hiload 26/600 Superdex 75 size-exclusion column connected to an ÄKTA Prime purification system, which had been washed previously with degassed 1 M NaOH and equilibrated with three column volumes of degassed standard purification buffer supplemented with 1 M

NaCl. Proteins were eluted using this buffer, and fractions containing βPGM were checked for purity using SDS-PAGE and pooled. βPGM$_{D10N}$ purifies readily from expression cultures as a highly stable *cis*-P βPGM$_{D10N}$:βG16BP complex[29]. Therefore, βPGM$_{D10N}$ was diluted into unfolding buffer (4 M guanidine hydrochloride, 50 mM K$^+$ HEPES (pH 7.2), 5 mM MgCl$_2$, 2 mM NaN$_3$), buffer-exchanged by dialysis into standard native buffer (50 mM K$^+$ HEPES (pH 7.2), 5 mM MgCl$_2$ and 2 mM NaN$_3$) and concentrated to 1 mM substrate-free βPGM$_{D10N}$ with a Vivapin (10 kDa molecular weight cut off, Sartorius) using a benchtop centrifuge operating at 3400 × *g* and 4 °C (Thermo Scientific Heraeus Labofuge 400 R). βPGM$_{D10N,P146A}$ purifies as substrate-free βPGM$_{D10N,P146A}$, therefore the unfolding-refolding step was omitted from the purification protocol. Protein concentrations were measured using a NanoDrop One$^C$ spectrophotometer (Thermo Scientific). Substrate-free βPGM samples were stored at –20 °C (βPGM molecular weight = 24.2 kDa, extinction coefficient = 19940 M$^{-1}$ cm$^{-1}$).

### Biosynthesis of βG16BP
βG16BP was produced enzymatically using the D170N variant of βPGM (βPGM$_{D170N}$)[41]. βG1P (20 mM) and acetyl phosphate (40 mM) were incubated with βPGM$_{D170N}$ (20 μM) in a 15 mL reaction volume containing 200 mM K$^+$ HEPES buffer, 100 mM MgCl$_2$ and 2 mM NaN$_3$ for 4 h at 25 °C. The reaction was quenched by heating at 90 °C for 10 min and the precipitated enzyme was pelleted using centrifugation (Sigma Model 3–15). The βG16BP-rich supernatant was filtered with a Vivaspin (5 kDa molecular weight cut off, Sartorius) using a benchtop centrifuge operating at 3900 × *g* and 4 °C (Thermo Scientific Heraeus Labofuge 400 R) and the resulting enzyme-free solution was passed through a 20 × 10 mm column packed with IR120 (H$^+$) ion-exchange resin that had been washed with 15 mL of milliQ water. The acidified flow-through was neutralised using 0.2 M barium hydroxide solution at 0 °C, which selectively precipitated βG16BP as a barium salt while maintaining both the βG1P and G6P barium salts in solution. The precipitate was pelleted using centrifugation at 3900 × *g* and 4 °C (Thermo Scientific Heraeus Labofuge 400 R) and the supernatant was discarded. The pellet was resolubilised in a large volume (~1 L) of cold milliQ water and passed through a 20 × 10 mm column packed with IR120 (Na$^+$) ion-exchange resin for conversion of βG16BP to the more soluble sodium salt. The flow through was then frozen at –80 °C and lyophilised to leave a fine powder as the final βG16BP product.

### F16BP
F16BP was obtained from Merck and was used without further purification. In solution, F16BP forms an equilibrium mixture of an α-anomer (15%), a β-anomer (81%) and two open chain forms with an interconversion rate of 8 s$^{-1}$[43]. The β-anomer of F16BP is the biologically active isomer.

### NMR spectroscopy
$^1$H$^{15}$N-TROSY NMR spectra of substrate-free $^{15}$N-labelled βPGM$_{D10N}$ and substrate-free $^{15}$N-labelled βPGM$_{D10N,P146A}$ were acquired at 298 K using a Bruker 600 MHz Neo spectrometer equipped with a 5-mm TCI cryoprobe and z-axis gradients. Samples contained 0.5 mM βPGM in standard NMR buffer (50 mM K$^+$ HEPES (pH 7.2), 5 mM MgCl$_2$, 2 mM NaN$_3$, 10% v/v $^2$H$_2$O and 1 mM trimethylsilyl propanoic acid (TSP)). Typically, $^1$H$^{15}$N-TROSY NMR spectra were accumulations of 32 transients with 256 increments and spectra widths of 32–36 ppm centred at 120 ppm in the indirect $^{15}$N-dimension. Experiments were processed using TopSpin4 (Bruker) and residue assignments were obtained by comparison with assigned $^1$H$^{15}$N-TROSY NMR spectra of substrate-free βPGM$_{WT}$ (BMRB 28095[13], BMRB 28096[13]) and substrate-free βPGM$_{P146A}$ (BMRB 27920[34]) using FELIX (Felix NMR, Inc.). Multi-dimensional heteronuclear NMR spectra for $^1$H, $^{15}$N and $^{13}$C backbone resonance assignment of the βPGM complexes were acquired at 298 K on a Bruker 800 MHz Neo spectrometer equipped with a 5-mm TCI cryoprobe and z-axis gradients. The standard suite of $^1$H$^{15}$N-TROSY and 3D TROSY-based constant time experiments were typically acquired (HNCA, HN(CO)CA, HNCACB, HN(CO)CACB, HN(CA)CO, HNCO, (H)N(COCA)NNH and H(NCOCA)NNH) using

non-uniform sampling employing a multi-dimensional Poisson Gap scheduling strategy with exponential weighting[44]. Non-uniform sampled data were reconstructed using either multi-dimensional decomposition or compressed sensing in TopSpin4[45]. $^1$H chemical shifts were referenced relative to the internal TSP signal resonating at 0.0 ppm, and $^{15}$N and $^{13}$C chemical shifts were referenced indirectly using nuclei-specific gyromagnetic ratios. Backbone resonance assignments for the βPGM complexes were obtained using FELIX (Felix NMR, Inc.) by comparing the correlated $^{13}$C chemical shifts of adjacent residues and were confirmed via the sequential backbone amide to amide correlations present in the (H)N(COCA)NNH and H(NCOCA)NNH NMR spectra. The populations of complexes present simultaneously in the NMR spectra were calculated using $^1$H$^{15}$N-TROSY peak intensities derived from a substantial number of residues. The βPGM$_{D10N}$:F16BP sample was generated using 0.5 mM substrate-free $^2$H$^{15}$N$^{13}$C-labelled βPGM$_{D10N}$ prepared in standard NMR buffer supplemented with 50 mM F16BP and 50 mM MgCl$_2$. Given the limited lifetime ( ~ 12 h), only $^1$H$^{15}$N-TROSY, HNCACB and HN(CA)CO spectra were acquired using two identical samples. The *cis*-P βPGM$_{D10N}$:F16BP complex (86% population and $^1$H, $^{15}$N and $^{13}$C assignments) and the *trans*-P βPGM$_{D10N}$:F16BP:MgT complex (14% population and $^1$H and $^{15}$N assignments) were present simultaneously in the NMR spectra. The βPGM$_{D10N,P146A}$:F16BP sample was generated using 0.5 mM substrate-free $^2$H$^{15}$N$^{13}$C-labelled βPGM$_{D10N,P146A}$ prepared in standard NMR buffer supplemented with 50 mM F16BP. The *trans*-A βPGM$_{D10N,P146A}$:F16BP:MgT complex (72% population and $^1$H, $^{15}$N and $^{13}$C assignments) and the *trans*-A βPGM$_{D10N,P146A}$:F16BP complex (28% population and $^1$H and $^{15}$N assignments) were present simultaneously in the NMR spectra. The elevated-MgCl$_2$ βPGM$_{D10N,P146A}$:βG16BP sample was generated using 0.5 mM substrate-free $^2$H$^{15}$N$^{13}$C-labelled βPGM$_{D10N,P146A}$ prepared in standard NMR buffer supplemented with 5 mM βG16BP and 100 mM MgCl$_2$. The *trans*-A βPGM$_{D10N,P146A}$:βG16BP:MgT complex (100% population and $^1$H, $^{15}$N and $^{13}$C assignments) was assigned as the only species in the NMR spectra. The βPGM$_{D10N,P146A}$:βG16BP sample was generated using 0.5 mM substrate-free $^2$H$^{15}$N$^{13}$C-labelled βPGM$_{D10N,P146A}$ prepared in standard NMR buffer supplemented with 5 mM βG16BP. The *trans*-A βPGM$_{D10N,P146A}$:βG16BP:MgT complex (44% population and $^1$H, $^{15}$N and $^{13}$C assignments), the *cis*-A βPGM$_{D10N,P146A}$:βG16BP complex (40% population and $^1$H, $^{15}$N and $^{13}$C assignments) and the *trans*-A βPGM$_{D10N,P146A}$:βG16BP complex (16% population and $^1$H and $^{15}$N assignments) were present simultaneously in the NMR spectra. Weighted chemical shift perturbations of the backbone amide group for each residue were calculated as: $\Delta\delta = [(\delta_{HN-X} - \delta_{HN-Y})^2 + (0.13 \times (\delta_{N-X} - \delta_{N-Y}))^2]^{1/2}$, where X and Y are the two complexes being compared.

## $^1$H$^{15}$N-TROSY chemical shift library

To facilitate backbone resonance assignment of the βPGM complexes, a $^1$H$^{15}$N-TROSY chemical shift library for each residue was constructed using the following assigned βPGM species and βPGM complexes: substrate-free *cis*-P βPGM$_{WT}$ (BMRB 28095[13]), substrate-free *trans*-P βPGM$_{WT}$ (BMRB 28096[13]), substrate-free *trans*-A βPGM$_{P146A}$ (BMRB 27920[34]), the *cis*-P βPGM$_{D10N}$:βG16BP complex (BMRB 27174[29]), the *cis*-P Mg$_{cat}$-free βPGM$_{D10N}$:βG16BP complex (BMRB 27175[29]), the *cis*-P βPGM$_{WT}$:MgF$_3$:G6P complex (BMRB 7234[26]) and the *cis*-A βPGM$_{P146A}$:MgF$_3$:G6P complex (BMRB 28097[13]). The $^1$HN (x-axis with reversed sense) and $^{15}$N (y-axis with reversed sense) chemical shifts were plotted for each residue, together with the $^1$H$^{15}$N-TROSY chemical shifts of substrate-free *cis*-P βPGM$_{D10N}$, substrate-free *trans*-P βPGM$_{D10N}$, substrate-free *trans*-A βPGM$_{D10N,P146A}$, the *cis*-P βPGM$_{D10N}$:F16BP complex (BMRB 51985), the *trans*-A βPGM$_{D10N,P146A}$:F16BP:MgT complex (BMRB 51986), the *trans*-A βPGM$_{D10N,P146A}$:F16BP complex (BMRB 51987), the *trans*-A βPGM$_{D10N,P146A}$:βG16BP:MgT complex (BMRB 51988), the *cis*-A βPGM$_{D10N,P146A}$:βG16BP complex (BMRB 51989) and the *trans*-A βPGM$_{D10N,P146A}$:βG16BP complex (BMRB 51990) (Supplementary Data 2, Supplementary Data 3). Analysis of the chemical shifts indicates that A143 and D180 are reporters of the

isomerisation state of the K145-X146 peptide bond and that I84 and S88 are reporters of the interdomain hinge closure angle.

## X-ray crystallography

Frozen aliquots of either substrate-free $^{15}$N-labelled βPGM$_{D10N}$ or substrate-free $^{15}$N-labelled βPGM$_{D10N,P146A}$ in standard native buffer (50 mM K$^+$ HEPES (pH 7.2), 5 mM MgCl$_2$ and 2 mM NaN$_3$) were thawed on ice and centrifuged briefly to pellet insoluble material. Crystals of the *cis*-P βPGM$_{D10N}$:F16BP complex were obtained from a solution of substrate-free βPGM$_{D10N}$ containing 50 mM F16BP. Crystals of substrate-free *trans*-A βPGM$_{D10N,P146A}$ were obtained from a solution of substrate-free βPGM$_{D10N,P146A}$ containing 3 mM AlCl$_3$, 20 mM NaF and 15 mM G6P. Crystals of the *trans*-A βPGM$_{D10N,P146A}$:F16BP:MgT complex were obtained from a solution of substrate-free βPGM$_{D10N,P146A}$ containing 50 mM F16BP and 50 mM MgCl$_2$. Crystals of the *trans*-A βPGM$_{D10N,P146A}$:βG16BP:MgT complex were obtained from a solution of substrate-free βPGM$_{D10N,P146A}$ containing 10 mM βG16BP and 50 mM MgCl$_2$. Solutions were adjusted to a final protein concentration of 0.6 mM, incubated for ~10 min and mixed 1:1 with precipitant (28–38% w/v PEG 4000), together with either 100 mM or 200 mM sodium acetate and either 100 mM or 200 mM tris-HCl (pH 7.2). Crystals were grown at 290 K by hanging-drop vapour diffusion using a 2 µL drop suspended on a siliconised glass coverslip above a 700 µL well. Small needle or rod-shaped crystals grew typically over several days. Crystals were harvested using a mounted LithoLoop (Molecular Dimensions Ltd.) and were cryo-protected in their mother liquor containing an additional 25% v/v ethylene glycol prior to plunging into liquid nitrogen. Diffraction data were collected at 100 K either on the i03 beamline (wavelength 0.9795 Å or 0.9801 Å) or the i04 beamline (wavelength 0.9795 Å or 0.9227 Å) at the Diamond Light Source (DLS), Oxfordshire, United Kingdom. Data were processed using the xia2 pipeline[46] and resolution cutoffs were applied using CC-half values. Structures were determined by molecular replacement using Phaser[47] within the CCP4 Cloud[48] using previously deposited βPGM structures with the most appropriate interdomain hinge closure angle (PDB 2WHE[26] or PDB 2WF9[27]) as a search model. Model building was achieved using COOT[49] and restrained refinement was performed with REFMAC5[50] in the CCP4i suite[51] with either isotropic temperature factors (resolutions >1.4 Å) or anisotropic temperature factors (resolutions <1.4 Å). Ligands were not included until the final stages of refinement to avoid biasing Fourier maps. Structure validation was carried out with COOT and MolProbity[52]. For the structure of substrate-free *trans*-A βPGM$_{D10N,P146A}$ (PDB 8Q1C), the MolProbity score is 0.97 (100$^{th}$ percentile, 1.68 ± 0.25 Å) and the Ramachandran statistics are: 97.0% favoured/allowed, 0.0% disallowed, 96.4% favoured rotamers and 0.0% poor rotamers. For the structure of the *cis*-P βPGM$_{D10N}$:F16BP complex (PDB 8Q1D), the MolProbity score is 0.74 (100$^{th}$ percentile, 1.75 ± 0.25 Å) and the Ramachandran statistics are: 97.3% favoured/allowed, 0.0% disallowed, 95.6% favoured rotamers and 0.6% poor rotamers. For the structure of the *trans*-A βPGM$_{D10N,P146A}$:F16BP:MgT complex (PDB 8Q1E), the MolProbity score is 0.69 (100$^{th}$ percentile, 1.23 ± 0.25 Å) and the Ramachandran statistics are: 98.2% favoured/allowed, 0.0% disallowed, 96.7% favoured rotamers and 0.0% poor rotamers. For the structure of the *trans*-A βPGM$_{D10N,P146A}$:βG16BP:MgT complex (PDB 8Q1F), the MolProbity score is 0.91 (100$^{th}$ percentile, 1.01 ± 0.25 Å) and the Ramachandran statistics are: 98.0% favoured/allowed, 0.0% disallowed, 97.4% favoured rotamers and 0.3% poor rotamers. Superpositions and crystallographic figures were prepared using PyMOL (The PyMOL Molecular Graphics System, version 1.8/2.2 Schrodinger LLC) and the interdomain hinge closure angle was calculated with DynDom[53].

## Calculation of intrinsic Euler angles

The changes in the cap and core interdomain relationship for the crystal structures, with respect to the *cis*-P βPGM$_{WT}$:MgF$_3$:G6P complex (PDB 2WF5[26]) reference structure, were described using intrinsic Euler angles. Under this framework, the reference structure was aligned to the principal axes derived from the positional distribution of Cα atoms in the cap domain,

thus setting the coordinate basis for the calculation of intrinsic rotations. Each crystal structure was first aligned to the cap domain of the reference structure, with subsequent alignment to the core domain. The rotation matrix associated with the transition from a cap-aligned structure to a core-aligned structure was used to calculate the three intrinsic Euler angles (pitch, roll and yaw). In this context, the pitch angle represents a cap and core closing angle, the roll angle represents a cap and core twisting motion and the yaw angle represents a cap and core left-to-right lateral rotation. Both the principal axes calculations and the structural alignments were implemented using MDAnalysis[54] and calculation of the intrinsic Euler angles from the rotation matrix was performed using SciPy[55].

## Statistics and reproducibility

Pearson correlation coefficients ($r$) were calculated in MATLAB R2021b using the *corrcoef* command. The correlation coefficients were transformed to Fisher z-transformed coefficients ($z_r = \frac{1}{2}\tanh^{-1} r$), and a two-sample z-test was performed to determine if the difference between independent correlation coefficients was statistically significant ($P$ value < 0.05).

## Reporting summary

Further information on research design is available in the Nature Portfolio Reporting Summary linked to this article.

## Data availability

Data supporting the findings of this manuscript are available from the corresponding author upon request. The atomic coordinates and structure factors have been deposited in the Protein Data Bank (www.rcsb.org) with the following codes: substrate-free *trans*-A βPGM$_{D10N,P146A}$ (PDB 8Q1C), the *cis*-P βPGM$_{D10N}$:F16BP complex (PDB 8Q1D), the *trans*-A βPGM$_{D10N,P146A}$:F16BP:MgT complex (PDB 8Q1E) and the *trans*-A βPGM$_{D10N,P146A}$:βG16BP:MgT complex (PDB 8Q1F). The NMR chemical shifts and associated time domain data have been deposited in the Bio-MagResBank (www.bmrb.wisc.edu) with the following accession numbers: the *cis*-P βPGM$_{D10N}$:F16BP complex (BMRB 51985), the *trans*-A βPGM$_{D10N,P146A}$:F16BP:MgT complex (BMRB 51986), the *trans*-A βPGM$_{D10N,P146A}$:F16BP complex (BMRB 51987), the *trans*-A βPGM$_{D10N,P146A}$:βG16BP:MgT complex (BMRB 51988), the *cis*-A βPGM$_{D10N,P146A}$:βG16BP complex (BMRB 51989) and the *trans*-A βPGM$_{D10N,P146A}$:βG16BP complex (BMRB 51990).

## Code availability

Code developed in Python for this study is publicly available and can be found on GitHub [https://github.com/adamjf15/PGM-Euler-Angles].

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

## Acknowledgements

We would like to thank Andrea Hounslow for support with the acquisition of heteronuclear NMR experiments. We would also like to thank the beamline scientists at the Diamond Light Source (DLS) for assistance with data collection and the Science and Technology Facilities Council (STFC) for funding and access to synchrotron radiation facilities (MX24447 and MX31850). This research was supported by Consejo Nacional de Ciencia y Tecnologia, Mexico (CONACYT; F.A.C.N., Grant Number 472448) and the Biotechnology and Biological Sciences Research Council (BBSRC; N.J.B., Grant Number BB/M021637/1 and BB/S007965/1).

## Author contributions

F.A.C.N., N.J.B. and J.P.W. designed research. F.A.C.N. produced isotopically enriched protein and βG16BP. F.A.C.N. acquired NMR experiments. F.A.C.N., N.J.B. and A.B. analysed NMR data. F.A.C.N., N.J.B. and P.J.B. performed and analysed X-ray crystallography experiments. A.J.F. performed the intrinsic Euler angle calculations. F.A.C.N., N.J.B., M.J.C. and J.P.W. wrote the paper with help from all the authors. All authors have given approval to the final version of the manuscript.

## Competing interests

The authors declare no competing interests.
