## [Transparent Peer Review file · Communications Biology]

Peri active site catalysis of proline isomerisation is the molecular basis of allomorphy in β -phosphoglucosyltransferase

Corresponding Author: Professor Jonathan Waltho

Version 0:

Reviewer comments:

Reviewer #1

(Remarks to the Author)

This study delves into the structural analysis of β -phosphoglucosyltransferase mutants when bound to two ligands exhibiting different activation properties. It reveals a cis-trans isomerization within the enzyme, wherein, depending on the ligand, it may transition into a fully active conformation known as NACIII. A key finding is the enzyme's self-catalysis of this cis-trans transition, while fructose 1,6-bisphosphate fails to induce the enzyme towards the active NACIII due to steric hindrance. The proposed catalysis of cis-isomerization of a Lys-Pro bond by the Lys side chain remains untested experimentally, tempering the enthusiasm for concluding that " β G16BP mediates catalysis of proline isomerization." Nevertheless, despite this concern, the publication of this work is warranted as the data enhance our comprehension of the allosteric regulatory mechanism of the enzyme. Moreover, the structural analyses conducted, employing both NMR and crystallography, demonstrate high quality.

As a suggestion, the authors may consider refining the manuscript for readability. Complex acronyms and abbreviations used to name structures could be simplified, and the presence of lengthy sentences could be reduced to enhance comprehension. Furthermore, it would be beneficial to condense the Discussion section to focus primarily on the novel structural insights provided by the manuscript. Specifically, the discussion surrounding Asp8 and Asp170 could benefit from a dedicated figure to aid comprehension, as it currently remains somewhat speculative and challenging to follow.

Reviewer #2

(Remarks to the Author)

This study discussed an allomorphic regulatory mechanism in β -phosphoglucosyltransferase (β PGM), which converts β -glucose 1-phosphate to glucose 6-phosphate. Full allomorphic activator β -glucose 1,6-bisphosphate prompts domain closure and the shift of enzyme populations between conformers, leading to rapid isomerization of a key proline residue. In contrast, partial activator fructose 1,6-bisphosphate slows phosphoryl transfer due to the lack of an adequate conformational shifting. The experimental evidence, based in x-ray structures and NMR measurements, supports the discussed mechanism. The paper is well-written in general. I just have a couple of suggestions to improve the manuscript:

- The authors still use the Near Attack Conformation (NAC) concept and nomenclature. This theory has been clearly overcome by detailed analysis of different contributions to catalysis. I recommend changing the nomenclature used in the manuscript. Otherwise, the authors do not really make an analysis in terms of reaction coordinates to classify the conformations as NACs.
- Based on the proximity of D8 and D170 side chains the authors propose that one of them must be protonated and that this proton could play a role during catalysis. It is difficult to judge this proposal without having access to the coordinates, but in principle the proximity of the magnesium ion seems to suggest that this would not be the case. It could be the case that one of these residues is protonated during intermediate, high-energy states of the reaction, but the suggestion of protonation in the resting state would deserve to be analyzed in more detail. Can the authors provide any evidence or examples of other enzymes where protonation of carboxylate groups belonging to the coordination of a magnesium ion has been suggested?
- It would be very useful for the reader to have a figure of the enzyme indicating the position of all the loops and other structural motifs that are discussed throughout the manuscript.

Reviewer #3

(Remarks to the Author)

This is a really excellent study in which NMR and crystallographic analysis are used to provide detailed insight into the allosteric regulation of β -phosphoglucomutase. The manuscript adds substantial detail to a prior report from the group (reference 31). The mechanism involves modulation of the cis/trans amide bond geometry of a key proline residue. I do think QM/MM type modelling would add to the current study, as also may (e.g.) HDX MS and solvent isotope studies. However, I do not think these are a requirement – I'd much prefer the authors do this carefully and present a detailed story as appears to be their modus operandi. Overall, I'm very supportive of the work and the authors should be congratulated on their excellent work.

I do think the title should mention the enzyme and the abstract should be rewritten, but these are the most critical things I can say. I'm keen for the authors to keep focussing on the details. A few things to consider below.

Main text specific points

Abstract

L34 – suggest 'Here we describe biophysical studies revealing the allomorphic regulations mechanism of β PGM'.

Maybe more precisely define the relevant domains.

L39 – 'twisted anti/endo proline amide....' Define endo more precisely (C4 endo?)

L105 – 'control of β PGM...'

L118 – 'anti/endo conformation for Pro... transition state'

The final paragraph of the introduction could be more concise/some of the detail should be in the abstract.

Results

Define the evidence/cite figure for the cis/trans %s

L140 - 'completely in the cis...?'

L181 – define which crystal/cite

L208 – define where the tris is

L218 – 'a structure'

Discussion

The roles of the (c4) endo/exo conformation in the isomerisation need to be more clearly introduced

Specific points - Supplementary Information

Very small points:

Fig S1 – negative charges/H OH OH could be 'aesthetically' improved

Fig S2 – labels could be bigger

Fig S3 – check the complexes mentioned in the title are clear

Fig S5/S9 – label N-/C-termini on top structures

Fig S7 - check title makes sense

Fig S11 – Z+ charge on the Mg?

Main text figures

Showing the 'chemical' structures of the substrates etc as currently in Fig S1, in main text figure 1 would be useful for the non-expert (also label N-/C-termini in figure 1C).

Figs 2/3 – make labels bigger please

Fig 6 – label N-/C-termini in top structure

Fig 7 – is the endo nature of the ring in 7b clear? / maybe label this. Maybe add 'transient' under b

Author Rebuttal letter:

We would like to thank the three reviewers for their insightful and helpful comments, which we feel have helped us to improve the clarity of the manuscript.

Reviewer #1 (Remarks to the Author):

This study delves into the structural analysis of β -phosphoglucomutase mutants when bound to two ligands exhibiting different activation properties. It reveals a cis-trans isomerization within the enzyme, wherein, depending on the ligand, it may transition into a fully active conformation known as NACIII. A key finding is the enzyme's self-catalysis of this cis-trans transition, while fructose 1,6-bisphosphate fails to induce the enzyme towards the active NACIII due to steric hindrance. The proposed catalysis of cis-isomerization of a Lys-Pro bond by the Lys side chain remains untested experimentally, tempering the enthusiasm for concluding that " ^{12}C G16BP mediates catalysis of proline isomerization." Nevertheless, despite this concern, the publication of this work is warranted as the data enhance our comprehension of the allosteric regulatory mechanism of the enzyme. Moreover, the

structural analyses conducted, employing both NMR and crystallography, demonstrate high quality.

As a suggestion, the authors may consider refining the manuscript for readability. Complex acronyms and abbreviations used to name structures could be simplified, and the presence of lengthy sentences could be reduced to enhance comprehension.

We accept that the names of the structures are lengthy and take some time to become familiar with, but we feel that it is best for us to stick with the naming protocols for this enzyme used in previous manuscripts. We have shortened lengthy sentences where we identified these.

Furthermore, it would be beneficial to condense the Discussion section to focus primarily on the novel structural insights provided by the manuscript. Specifically, the discussion surrounding Asp8 and Asp170 could benefit from a dedicated figure to aid comprehension, as it currently remains somewhat speculative and challenging to follow.

In light of this comment and a related comment from Reviewer #2, we have decided to condense the Discussion section. We have moved the discussion surrounding Asp8 and Asp170 to the Supplementary Information. We accept that the proposed model remains unproven and was tangential to the main focus of the Discussion, but we felt that the unusual proximity of the two carboxylate groups in the cis-P $\hat{\text{P}}\text{GMD10N:F16BP}$ complex crystal structure (PDB 8Q1D) was worthy of comment. A dedicated figure (Supplementary Figure 3) is also included in the Supplementary Information to aid comprehension of our proposed model.

Reviewer #2 (Remarks to the Author):

This study discussed an allomorphic regulatory mechanism in $\hat{\text{P}}^2$ -phosphoglucomutase ($\hat{\text{P}}^2\text{PGM}$), which converts $\hat{\text{P}}^2$ -glucose 1-phosphate to glucose 6-phosphate. Full allomorphic activator $\hat{\text{P}}^2$ -glucose 1,6-bisphosphate prompts domain closure and the shift of enzyme populations between conformers, leading to rapid isomerization of a key proline residue. In contrast, partial activator fructose 1,6-bisphosphate slows phosphoryl transfer due to the lack of an adequate conformational shifting. The experimental evidence, based in x-ray structures and NMR measurements, supports the discussed mechanism. The paper is well-written in general. I just have a couple of suggestions to improve the manuscript:

- The authors still use the Near Attack Conformation (NAC) concept and nomenclature. This theory has been clearly overcome by detailed analysis of different contributions to catalysis. I recommend changing the nomenclature used in the manuscript. Otherwise, the authors do not really make an analysis in terms of reaction coordinates to classify the conformations as NACs.

We simply use this nomenclature as a convenient way of labelling closed structures as they all adhere to the original geometric definitions of NACs. We were not trying to imply a specific model or relationship to reaction coordinate with this nomenclature. For consistency with considerable previous literature on this enzyme, though, we feel that it is best to retain this nomenclature.

- Based on the proximity of D8 and D170 side chains the authors propose that one of them must be protonated and that this proton could play a role during catalysis. It is difficult to judge this proposal without having access to the coordinates, but in principle the proximity of the magnesium ion seems to suggest that this would not be the case. It could be the case that one of these residues is protonated during intermediate, high-energy states of the reaction, but the suggestion of protonation in the resting state would deserve to be analyzed in more detail. Can the authors provide any evidence or examples of other enzymes where protonation of carboxylate groups belonging to the coordination of a magnesium ion has been suggested?

As included in the response to Reviewer #1, we have decided to move this whole Discussion paragraph to the Supplementary Information.

- It would be very useful for the reader to have a figure of the enzyme indicating the position of all the loops and other structural motifs that are discussed throughout the manuscript.

We have prepared a new manuscript figure (Figure 2) that highlights the structural architecture of β -PGM. The positions of the key functional loops and structural motifs have been labelled accordingly.

Reviewer #3 (Remarks to the Author):

This is a really excellent study in which NMR and crystallographic analysis are used to provide detailed insight into the allosteric regulation of β -phosphoglucomutase. The manuscript adds substantial detail to a prior report from the group (reference 31). The mechanism involves modulation of the cis/trans amide bond geometry of a key proline residue. I do think QM/MM type modelling would add to the current study, as also may (e.g.) HDX MS and solvent isotope studies. However, I do not think these are a requirement - I much prefer the authors do this carefully and present a detailed story as appears to be their modus operandi. Overall, I am very supportive of the work and the authors should be congratulated on their excellent work.

I do think the title should mention the enzyme and the abstract should be rewritten, but these are the most critical things I can say. I am keen for the authors to keep focussing on the details. A few things to consider below.

We have added the name of the enzyme to the Title.

Main text specific points

Abstract

L34 - suggest - Here we describe biophysical studies revealing the allomorphic regulations mechanism of β -PGM.

Maybe more precisely define the relevant domains.

L39 - twisted anti/endo proline amide - Define endo more precisely (C4 endo?)

We have rewritten the Abstract to take these comments into account and to shorten the Abstract in line with the Journal recommendations.

L105 - control of β -PGM -

L118 - anti/endo conformation for Pro - transition state

The final paragraph of the introduction could be more concise/some of the detail should be in the abstract.

We have reworded the Introduction to take into account these comments. The final paragraph has been shortened and the Abstract changed, subject to the limitations in the length of the latter.

Results

Define the evidence/cite figure for the cis/trans %s

We have moved Fig. 5 to this point in the manuscript to illustrate that both cis-P and trans-P substrate-free β -PGMWT and β -PGMD10N are present in solution. The figures have been renumbered throughout to reflect this change. New text has been inserted (L124-L130) to indicate how the relative populations of species present simultaneously in solution were calculated:

On removal of β -G16BP, substrate-free β -PGMD10N in standard NMR buffer is present in solution as a mixed population of cis-P β -PGMD10N (58 %) and trans-P β -PGMD10N (42 %) species that both adopt an open conformation (Fig. 3). This behaviour mirrors that of substrate-free wild-type β -PGM under the same conditions (cis-P β -PGMWT (66 %), BMRB

2809513 and trans-P $\hat{\text{I}}^2\text{PGMWT}$ (34 %), BMRB 2809613). The relative populations of species present simultaneously in solution were calculated using $^1\text{H}^{15}\text{N}$ -TROSY peak intensities derived from a substantial number of residues.

L140 - completely in the cis?

We have amended the text accordingly.

L181 - define which crystal/cite

We have added this detail to the text.

L208 - define where the tris is

We have rewritten the sentence to include this detail, as follows:

In each active site, an inorganic phosphate anion is coordinated by the sidechains of K117 and R49, along with a tris molecule (derived from the crystallisation buffer) that occupies a similar location to the sugar ring of the allomorphic activators (Supplementary Fig. 2a, b).

L218 - structure

The two occurrences of this wording in the manuscript have been changed accordingly.

Discussion

The roles of the (c4) endo/exo conformation in the isomerisation need to be more clearly introduced

We have modified the text that introduces the proline isomerisation model (L407-L411), and the C4-endo nature of the proline ring has been denoted throughout the manuscript.
Specific points - Supplementary Information

Very small points:

Fig S1 - negative charges/H OH OH could be aesthetically improved

We have amended the positions of negative charges on the phosphate groups present in the active site and moved the H atom of $\hat{\text{I}}^2\text{G6P}/\hat{\text{I}}^2\text{G1P}$ to improve the aesthetic appearance of the figure.

Fig S2 - labels could be bigger

We have increased the size of the labels in the figure.

Fig S3 - check the complexes mentioned in the title are clear

We have amended the caption to include all the substrate-free $\hat{\text{I}}^2\text{PGM}$ species and $\hat{\text{I}}^2\text{PGM}$ complexes mentioned in the title.

Fig S5/S9 - label N-/C-termini on top structures

We have added N- and C-termini labels to the top structure in the figure.

Fig S7 - check title makes sense

We have amended the title to improve clarity.

Fig S11 - Z+ charge on the Mg?

We have added 2+ charges to the Mg atoms in the figure.

Main text figures

Showing the chemical structures of the substrates etc as currently in Fig S1, in main text figure 1 would be useful for the non-expert (also label N-/C-termini in figure 1C).

We have included the chemical structures for ^{18}F 16BP and ^{18}G 16BP in the figure and also added N- and C-termini labels to the ^{18}P GM structural overlays.

Figs 2/3 - make labels bigger please

We have increased the size of the labels in these figures.

Fig 6 - label N-/C-termini in top structure

We have added N- and C-termini labels to the top structure in the figure.

Fig 7 - is the endo nature of the ring in 7b clear? / maybe label this. Maybe add "transient" under b

We have improved the clarity of the C4-endo nature of the proline ring in the figure by slightly changing the perspective of the bonds and inserting a label. However, we felt that the inclusion of the term transient within the figure to describe the model of the transition state of proline isomerisation (indicated with square brackets and a double dagger) was difficult to fit in the space. Instead, we have added this detail to the figure caption.

Version 1:

Reviewer comments:

Reviewer #1

(Remarks to the Author)

The Authors have addressed most of the comments by the Reviewers. The clarity of the manuscript has improved though it arguably remains somewhat difficult for the lay reader, not working on this field.

Reviewer #2

(Remarks to the Author)

The authors have addressed sufficiently the points raised in my previous report. In my opinion the paper can be accepted for publication.

Reviewer #3

(Remarks to the Author)

The authors have been good efforts to address the issues raised by myself and the other reviewers. Congratulations on a nice study.
